



# NEE estimates 2006-2019 over Europe from a pre-operational ensemble-inversion system

Saqr Munassar[1], Christian Rödenbeck[1], Thomas Koch[1,2], Kai U. Totsche[3], Michał Gałkowski[1,4], Sophia Walther[1], Christoph Gerbig[1]

[1]Max-Planck Institute for Biogeochemistry, Jena, Germany
[2]Meteorological Observatory Hohenpeissenberg, Deutscher Wetterdienst, Hohenpeissenberg, Germany
[3]Institute of Geoscience, Friedrich Schiller University, Jena, Germany
[4]AGH University of Science and Technology, Kraków, Poland

*Correspondence to*: Saqr Munassar (smunas@bgc-jena.mpg.de)

**Abstract.** 3-hourly Net Ecosystem Exchange (NEE) is estimated at spatial scales of 0.25 degrees over the European continent, based on the pre-operational inverse modelling framework "CarboScope Regional" (CSR) for the years 2006 to 2019. To assess the uncertainty originating from the choice of a-priori flux models and observational data, ensembles of inversions were produced using three terrestrial ecosystem flux models, two ocean flux models, and three sets of atmospheric stations. We find that the station set ensemble accounts for 61% of the total spread of the annually aggregated fluxes over the full domain when varying all these elements, while the biosphere and ocean ensembles resulted in much smaller contributions to the spread of 28% and 11%, respectively. These percentages differ over the specific regions of Europe, based on the availability of atmospheric data. For example, the spread of the biosphere ensemble is prone to be larger in regions that are less constrained by $CO_2$ measurements. We further investigate the unprecedented increase of temperature and simultaneous reduction of Soil Water Content (SWC) observed in 2018 and 2019. We find that NEE estimates during these two years suggest an impact of drought occurrences represented by the reduction of Net Primary Productivity (NPP), which in turn lead to less $CO_2$ uptake across Europe in 2018 and 2019, resulting in anomalies up to 0.13 and 0.07 PgC yr$^{-1}$ above the climatological mean, respectively. Annual temperature anomalies also exceeded the climatological mean by 0.46 °C in 2018 and by 0.56 °C in 2019, while standardized-precipitation-evaporation-index (SPEI) anomalies declined to -0.20 and -0.05 SPEI units below the climatological mean in both 2018 and 2019, respectively. Therefore, the biogenic fluxes showed a weaker sink of $CO_2$ in both 2018 and 2019 (-0.22±0.05 and -0.28±0.06 PgC yr$^{-1}$, respectively) in comparison with the mean -0.36±0.07 PgC yr$^{-1}$ calculated over the full analysed period (i.e., fourteen years). These translate into a continental-wide reduction of the annual sink by 39 % and 22 %, respectively, larger than the typical year-to-year standard deviation of 19 % observed over the full period.



## 1 Introduction

The atmospheric mole fractions of Greenhouse Gases (GHGs) like $CO_2$, $CH_4$, and $N_2O$ have drastically increased since the industrial era began (Friedlingstein et al., 2019), primarily caused by anthropogenic GHG emissions. As a consequence, the globally averaged surface air temperature has risen by 0.87 °C from 1850 to 2015 (Jia et al., 2019). Carbon dioxide is ranked as the most prominent anthropogenic GHG owing to its atmospheric abundance, resulting from a) the natural exchange through the biogeochemical interactions with the organic molecules in the biosphere and hydrosphere (represented by the Net Primary

Productivity - NPP), b) significant anthropogenic emissions from burning of fossil carbon and from cement production, and c) land use changes such as deforestation. The largest uptake of atmospheric $CO_2$ is carried out through terrestrial Gross Primary Production (GPP) and thought to derive an uptake of about one-third of anthropogenic emissions owing to enhancement of photosynthetic $CO_2$ uptake in the recent decades (Cai and Prentice, 2020). However, measurements of NEE cannot be easily achieved at finer spatial and temporal scales over the globe. Ancillary data from the atmosphere and the biosphere are thus

applied in the inverse modelling setups to estimate the natural $CO_2$ fluxes. Such a method of using atmospheric data to constrain NEE obtained from the terrestrial biogenic models is also called a top-down method.

The continuous expansion of GHG in-situ measurement capabilities enabled atmospheric tracer inversion systems to better infer the sources and sinks of $CO_2$ at global (Ciais et al., 2010; Enting et al., 1995; Kaminski et al., 1999; Rödenbeck et al., 2003), and regional scales (Gerbig et al., 2003; Kountouris et al., 2018b; Lauvaux et al., 2016). Meanwhile, regional

atmospheric inversions have employed atmospheric transport models at finer spatial resolution to deal with the complex atmospheric circulation at continental measurement stations.

The observational site network across Europe has been markedly homogenized since the Integrated Carbon Observation System (ICOS) was established in 2015, allowing for better estimation of the regional budgets of $CO_2$ over Europe (Monteil et al., 2020). Consequently, this has allowed for a better understanding of the impacts of climate extremes on the ecosystem

productivity such as the drought episode that occurred in 2018 (Bastos et al., 2020; Rödenbeck et al., 2020; Thompson et al., 2020). The inversions typically assume anthropogenic emissions to be well known, and thus target the more uncertain biosphere-atmosphere fluxes.

The regional inversion framework encounters various sources of errors, such as 1) the uncertainty of a-priori knowledge (necessary in the Bayesian framework inversions to regularize the solution of the ill-posed inverse problem), and 2) the

representation error resulting from the inaccuracies in simulating the atmospheric transport. The structure of prior error (e.g., uncertainties in the prior biosphere flux estimates) is of particular importance as it determines the way in which the flux corrections calculated from the data information should be spread in space and time (Chevallier et al., 2012; Kountouris et al., 2015). Defining proper error covariance matrices in both flux and measurement space is therefore essential to obtain an optimal estimate of the true fluxes. Non-optimized flux components used as prescribed fluxes in the inverse frameworks should be

provided with the highest achievable confidence, as any error in these components will directly modify the estimated biosphere-atmosphere fluxes.





Here, we present NEE estimates from a pre-operational regional inversion system set up over Europe covering fourteen years since 2006. An ensemble is created by varying a) a-priori biogenic fluxes, b) a-priori ocean fluxes, and c) the number of available atmospheric observation sites in order to estimate their impact on a-posteriori optimized biogenic fluxes. We
furthermore discuss the interannual variability (IAV) over this period, with special focus on the changes of NEE in 2018 and 2019, specifically in light of the water availability and temperature variations that occurred in the wake of anomalously warm and dry conditions over the continent. These changes are analysed using the seasonal and annual NEE fluxes aggregated over different subregions in Europe.

The inversion setup, observational dataset, and prior fluxes used are described in Section 2, including details on ensemble
members configuration. A statistical analysis of uncertainty and spreads over the ensembles of inversions are presented and discussed in Section 3.1. Section 3.2 presents the NEE estimated in the pre-operational inverse system based on several analysed cases. Finally, discussions and conclusions are summarized in Section 4.

## 2 Methods

### 2.1 Inversion framework

The CarboScope Regional inversion system (CSR) is used to infer NEE from observed atmospheric $CO_2$ dry mole fractions at high spatiotemporal resolution over Europe. The CSR makes use of Bayesian inference to regularise the solution of the under-determined inverse problem (i.e., there are more unknown fluxes than atmospheric observations). For details about the mathematical concepts, we refer the reader to Rödenbeck (2005); the specifications of the set-up largely follow previous studies by Kountouris et al. (2018a, 2018b). The inversion at the regional domain is embedded into the global atmosphere using the
"two-step scheme" described in Rödenbeck et al. (2009).

Atmospheric transport is simulated by the Stochastic Time-Inverted Lagrangian Transport (STILT) model (Lin et al., 2003), which is utilized to calculate hourly surface influences at the stations (i.e., "footprints") at the spatial resolution of 0.25° (longitude) × 0.25° (latitude). The model is driven by meteorological fields from the high-resolution implementation of Integrated Forecasting System (IFS HRES) model of the European Centre for Medium Range Weather Forecasts (ECMWF).
The upstream influence is simulated over the past 10 days by releasing 100 virtual particles at the sampling heights of stations (receptors). Additionally, we use the Eulerian global model TM3 (Heimann and Körner, 2003) within the CarboScope global inversion framework (Rödenbeck et al., 2003) to provide the far field contributions to the regional domain at a coarser spatial resolution of 5° (longitude) × 4° (latitude).

### 2.2 Atmospheric data

Since $CO_2$-dry-mole-fractions are the main constraint of the inversion system, we have aimed in our study to maximize the data coverage by using the observations available through the ICOS network as well as further atmospheric observation sites (both ICOS-associated and independent). All of the datasets are high-quality products of the level 2 ICOS Atmospheric


Release, which underwent a strict filtration procedure described in Hazan et al. (2016). This homogenized data treatment makes the data suitable for inverse modelling. For measurement sites with multiple sampling levels, the top one is chosen, as this one

is expected to be represented best in the STILT transport model.

The core of our observational dataset consists of data from 44 sites collected in the ICOS Carbon Portal under the 2018 drought initiative (https://doi.org/10.18160/ERE9-9D85), covering the period 2006-2018. This base dataset was extended into 2019 by the level 2 data (L2) released by ICOS in 2020, as well as included data from four new sites. From the total number of sites, 23 are currently ICOS-labelled and provide data since 2015, while the rest are non-labelled sites, providing datasets since

2006. Figure 1 shows the distribution of all sites throughout the domain of Europe. The figure also shows the division of the domain into six sub-regions (North, South, West, East, South-east, and Central Europe) used for post-processing, to outline the impact of the observational constraint distribution on posterior fluxes.

South-eastern Europe (light red).

The representation error is assumed to be specific for different station types, which are categorized to 5 classes. Weekly

representation errors are presented in Table 1, defining the measurement error covariance matrix in the cost function. The observations are mostly provided at hourly frequency, especially in recent years. We also include measurements from flask sampling (mostly weekly) when available from the corresponding sites. To better represent the well-mixed boundary layer in the STILT model, we limit our analyses to measurements of 6-hour day-time for all stations, i.e., 11:00 to 16:00 (local time), except for mountain sites. For the latter, night-time hours (23:00 to 04:00, local time), are chosen, during which these sites

regularly sample the residual layer. Particularly before establishing ICOS in 2015, the variability of station data coverage across the period of interest was rather high, underlining the sparsity of available data (Fig. 2). Since then, the site network over Europe has been markedly expanded as new stations were installed. It should be noted that the variability in data coverage is expected to result in an inconsistency of annual flux variations. Therefore, we combined stations into 3 subsets: a) the full set of stations, referred to as "all sites", b) a subset of 15 stations that have consistent coverage from 2006 to 2019, to which

we will refer as "core sites"), and c) a third subset of 16 stations that do not have gaps longer than 1 month during 2015-2019, subsequently referred to as "recent sites". The far field contributions provided to the regional domain were calculated in the two-step inversion approach (Rödenbeck et al., 2009) using a global observational record from 75 sites (doi:10.17871/CarboScope-s10oc_v2020), which has best data coverage in the 2010-2019 period.

## 2.3    A-priori-fluxes

Terrestrial ecosystem flux models are utilized to provide prior knowledge of biogenic fluxes (NEE, defined as the net ecosystem exchange). To appropriately represent the diurnal cycle in our modelling framework, NEE is obtained from the biosphere models at hourly temporal resolution. Three biosphere models were used as priors in the inversion runs. The first is the Vegetation Photosynthesis and Respiration Model, VPRM (Mahadevan et al., 2008). VPRM is a diagnostic model driven by shortwave radiation and temperature at 2-meter from the ECMWFs high resolution operational forecast product (IFS

HRES). To calculate NEE and respiration fluxes, it uses MODIS (Moderate Resolution Imaging Spectroradiometer) indices



derived from surface reflectance, namely Enhanced Vegetation Index (EVI) and land surface water (LSWI) together with type-specific vegetation parameters optimized against the eddy covariance (EC) data. Parameter values for VPRM previously used by Kountouris et al. (2018b), were updated using the most recent EC data, and are available at https://www.bgc-jena.mpg.de/bsi/index.php/Services/VPRMparam. The second biosphere prior is from the data driven modelling approach

FLUXCOM, which combines eddy covariance measurements and satellite observations in several machine learning algorithms to quantify the surface-atmosphere energy and carbon fluxes (Jung et al., 2020). Here, we use an extension of the modelling set-up described in Bodesheim et al. (2018), which employs daily and hourly surface meteorological information from ERA5 as well as a mean annual cycle of satellite data to produce NEE estimates at hourly temporal and 0.5-degree spatial resolution. It should be noted that the magnitude of interannual changes in the data-driven flux estimates is generally found to be

unrealistically small (Jung et al., 2012). The terrestrial biosphere model SiBCASA (Schaefer et al., 2008) is used as the third biosphere model. SiBCASA is a combined framework based on the Simple Biosphere (SiB) model and the Carnegie-Ames Stanford Approach (CASA) model. As explained in Schaefer et al. (2008), Gross Primary Productivity (GPP) is calculated by SiB assuming that it is in balance with heterotrophic and autotrophic respiration ($R_H$, $R_A$), meaning that diurnal and seasonal variations are well represented; however, the long-term terrestrial carbon changes cannot be predicted by SiB component alone.

CASA fills this gap as it includes a light-use-efficiency model to estimate Net Primary Productivity (NPP, equal to GPP - $R_A$). In turn, CASA cannot calculate NEE during night time. Therefore, combining both models in a hybrid version of SiBCASA combines the advantages in their biophysical and biogeochemical aspects to calculate NEE from $R_H$, $R_A$ (SiB) and NPP (CASA).

Following Kountouris et al. (2018b), we assume that the spatial correlation of the prior uncertainty follows a hyperbolic decay

function, similar as in the inversion case nBVH described in that study. One notable difference in the current work is the improved implementation of the directional dependence, with a twofold increase of decay distance in meridional direction. Temporarily, prior errors are assumed to be correlated over 30 days, as found in Kountouris et al. (2015).

Ocean $CO_2$ fluxes and anthropogenic emissions are considered as prescribed fluxes in the inversion system. Ocean fluxes are taken from two sources at coarse spatial resolution, (5 × 4 degrees): a climatological flux product with monthly fluxes of

Mikaloff Fletcher et al. (2007), and the CarboScope $pCO_2$-based ocean flux, providing fluxes at 6-hourly temporal resolution (Rödenbeck et al., 2013). Fossil fuel emissions are taken from the category and fuel-type specific emission inventories of EDGAR-v4.3, and processed following the COFFEE approach (Steinbach et al., 2011) to include diurnal, weekly , seasonal, and annual variations. These emissions are updated annually according to national consumption data from the BP (British Petroleum) statistical review of world energy (BP, 2019), and are available from the ICOS Carbon Portal under

https://doi.org/10.18160/Y9QV-S113.

## 2.4     Set-up of the inversion runs

We conduct three ensembles of inversion runs listed in Table 2 utilizing different setups of prior products (biosphere and ocean ensembles), as well as selected sets of observational data (station set ensemble). The inversion runs are labelled with unique





codes for reference. B0 is defined as the base case of our analysis. It is configured using default settings of the inversion runs,
with biogenic fluxes from VPRM, climatological ocean fluxes, and using all available atmospheric data as input. In the
biosphere ensemble (consisting of B0, B1, B2), FLUXCOM and SiBCASA replace the VPRM model in both B1 and B2,
respectively, allowing for distinguishing the effect of using different prior flux models on posterior NEE. In the ocean
ensemble, we replace the climatological ocean fluxes used in B0 with the pCO$_2$-based CarboScope ocean fluxes in O1. The
station set ensemble is formed by running the inversion with varying measurement station subsets: B0 - all sites, S1 - core
sites, and S2 - recent sites, as explained in Section 2.2. For each of the three ensembles of inversions, its spread is calculated
as the standard deviation of the differences between each ensemble member and the ensemble mean over the respective
overlapping period of time. The statistical uncertainty calculations are calculated in the inverse system based on model-data
mismatch, performed for the base case inversion (B0) as they remain identical independent of the biosphere model used.

## 3 Results

### 3.1 Statistical analysis of ensemble uncertainties

The annual NEE estimates among the biosphere ensemble (Fig. 3) show good agreement across the three biosphere models
but also across S1 and O1 inversions, yielding similar budgets of CO$_2$ fluxes over the full domain. The findings suggest that
atmospheric data constraints are more dominated in the posterior NEE fluxes in comparison with the prior constraint.
Subregions that are characterized by strong observational constraints such as Central Europe show a closer consistency in the
posterior fluxes despite large prior differences among the biosphere models compared to the regions less constrained by
atmospheric data, such as Northern Europe.

Noteworthy, there is a striking similarity of interannual variations between the a-posteriori fluxes and both VPRM and
SiBCASA prior fluxes for the years 2009-2013. This agreement does not necessarily mean that posterior interannual variability
(IAV) is driven by biosphere models. This can be deduced from B1 (FLUXCOM) estimates of which the IAV differ in both
the prior and posterior fluxes, and where FLUXCOM NEE has weak interannual variations. Instead, VPRM and SiBCASA
are likely to entail this signal from the meteorological data used to force these models. However, the VPRM model
overestimates the mean CO$_2$ uptake compared to the a-posteriori fluxes, while SiBCASA underestimates the mean CO$_2$ uptake,
and this dissimilarity is persistent for all years as well.

The statistical uncertainty and spreads over the ensembles are evaluated, and affect our data (Fig.3). It is noticed that the
spreads over the posterior fluxes and prior fluxes are comparable with the corresponding uncertainties over the full domain
(All Europe), Central Europe, and Northern Europe. There is a clear reduction of uncertainty and spread in posterior fluxes
either over the full domain (All Europe) or in subregions like Central and Northern Europe. Unlike prior uncertainty, posterior
uncertainty slightly differs from year to year following the number of atmospheric sites available (Fig 2). This gets even clearer
when looking at the marked reduction of the posterior uncertainty in Central Europe, as well as in regions with high station
density, resulting in a stronger observational constraint. In contrast to Central Europe, a smaller reduction of posterior spread



is found in Northern Europe as well as in other regions where there are few or no stations (e.g., Eastern Europe, not shown). In this case, NEE estimates are not well constrained by atmospheric data. Instead, a-posteriori flux is driven by the inversion using biosphere models and their uncertainty, particularly for the distant areas that cannot be constrained by observations through the spatial correlation. Table 3 denotes the reduction of the biosphere ensemble spread in the a-posteriori relative to

the a-priori over the full domain, Central and Northern Europe. It indicates less reduction in Northern Europe due to the sparseness of observational sites. The large reduction of spread in Central Europe reflects a notable dependency of NEE estimates on the atmospheric measurements, substantially where the observation network is dense.

To analyse the seasonal variations, seasonal cycle from B0, B1, B2, S1, and O1 inversions is averaged over 13 years for the full domain of Europe together with the corresponding biogenic prior fluxes for VPRM, FLUXCOM, and SiBCASA (Fig.4).

Results show good agreement amid a-posteriori results of all inversions, while prior biosphere models show large differences, a pattern similar to the one seen over the annual fluxes in Fig.3. Nevertheless, posterior NEE fluxes estimated in S1 inversion show differences during May-August when compared with the estimates of other runs, reflecting a larger sensitivity of IAV to summer fluxes when applying a different set of stations. In addition, the difference of posterior fluxes seen in Fig. 3 over the annually aggregated estimates computed from B1 inversion over the period 2014-2018 largely results from the estimates during

May and June when comparing it to the rest of biosphere ensemble elements (Fig. 4).

Figure 5 illustrates the statistical uncertainty and the spread through the overall ensembles of inversions (listed in Table 2) calculated annually over three regions. As was discussed in the time series of NEE (Fig. 3), a reduction in posterior NEE uncertainty with respect to the assumed prior error is clear (dark grey bars in Fig. 5). A larger reduction is realized in Central Europe, emphasising a strong atmospheric signal constraint in the inversion. The spread among ensemble members (Fig. 5,

yellow bars) represents the standard deviation of the respective inversion results. The ensemble spread over the a-priori biosphere models agrees with the assumed prior uncertainty, with relatively high value (about 0.44 PgC yr$^{-1}$ domain-wide) indicating large discrepancies between prior flux models. This confirms that the prior uncertainty assumed in the CSR system is realistic. The IAV of B0 was calculated separately for prior and posterior fluxes (blue bars) from the anomalies relative to the long-term mean to reveal the magnitudes of interannual deviation in comparison with the spread variability.

In terms of the spread of the biosphere ensemble, the standard deviation of posterior fluxes declines from 0.666 to 0.032 PgC yr$^{-1}$ over All Europe. This reduction is 95.1%, 96.0%, and 74.8% in All Europe, Central Europe, and Northern Europe, respectively. Spatial differences are expected as stations are not evenly distributed across the domain of Europe. This can be noticed from the spread over Central Europe (a large number of stations, 18 sites) and Northern Europe. (a lesser number of stations, 8 sites). As a result, lack of observations leads to inflating the spread over the biosphere ensembles.

The largest impact on NEE estimates in the ensembles are observed when the spread over station set ensemble is analysed. In this regard, a robust analysis can be based on a subset from Central Europe, as the subsets of stations in this region are clearly contrasting in the two ensemble members (core sites and recent sites). The spread of the station set ensemble was found to be 0.11 PgC yr$^{-1}$ - larger than those resulting from the biosphere and ocean ensembles (0.05 and 0.02 PgC yr$^{-1}$, respectively). Noteworthy, the spread in the station set ensemble is slightly larger than the statistical uncertainty, highlighting the importance





of performing ensembles of inversions using different numbers of stations to assess the posterior uncertainty. In addition, NEE estimated among the station set ensemble suggests that while in all cases the posterior fluxes are data-driven, modification of the observation inputs leads to interannual variations. The spread of the ocean ensemble remains the smallest in all the regions, pointing out quite a weak dependency of posterior NEE estimates on ocean fluxes, in particular over inland regions (e.g., Central Europe).

The spatially distributed spread of all ensembles is depicted in Fig. 6. In this instance, the standard deviation was calculated for each grid cell rather than aggregating fluxes over regions first and then computing the spread (Fig. 5). The spatial spread here illustrates the deviations of the biosphere ensemble (Biosphere spread), the biosphere models (Prior biosphere spread), the station set ensemble (Stationset spread), and the ocean ensemble (Ocean spread). The maximum spread of $0.191 \times 10^{-4}$ (PgC yr$^{-1}$) was observed over the a-priori terrestrial biosphere models, particularly concentrated in Central and Southern Europe.

The spread of a-posteriori biosphere ensemble is significantly reduced. In the station set ensemble, isolated stations like Hegyhatsal in Hungary and Sierra de Gredos in Spain demonstrate relatively high impact on the NEE spatial patterns over broader areas, reflecting the inversion correlation length. However, such an impact is not clearly realized on the "Stationset spread" map amid dense clusters of sites due to the commutative constraint that compensates for the excluded sites within the subsets of stations. These results highlight the importance of defining a proper function of spatial correlation decay in the prior

error structure. A quite small influence is seen through the spread over ocean where a slight impact emerges only in wider coastal regions, being almost negligible inland (e.g., in Central and Eastern Europe).

Figure 7 indicates the spatial distributions of prior and posterior NEE averaged over the full 13-year period, estimated from B0, B1, and B2 inversions, as well as the corresponding innovation of fluxes (the difference between posterior and prior fluxes). Positive corrections have been made to the biosphere flux models that are regarded to be negatively biased (VPRM

and FLUXCOM, as was unequivocally confirmed by the annual time series of NEE in Fig. 3). In contrast, SiBCASA results are closer to the mean of posterior fluxes, with a small domain-wide negative correction, except for local positive innovations seen over Northern Germany and Western Mediterranean coast.

### 3.2    NEE estimates of 2018 and 2019 in a preoperational system

In this section, we present $CO_2$ fluxes for two selected years estimated in a preoperational system in the context of long-term

estimates. The period of interest is chosen to start from 2006 in which a better coverage of observations exists within the domain of Europe. Here, we give special attention to analysis of the drivers of spatiotemporal differences in line with climate disturbances that occurred in 2018 and 2019, during which inaccuracies of estimating the continental fluxes of $CO_2$ have been reported (Friedlingstein et al., 2019). This is due to the sensitivity of ecosystem respiration and photosynthetic fluxes to extreme events like lasting droughts. The analysis here is based on two inversion runs using observational data only from the subset of

core sites that have consistent measurements, i.e., as used in scenario S1 discussed in previous sections. The choice of using consistent measurements is essential to study the IAV to diminish the uncertainty caused by gaps of data coverage over years.





The IAV of estimated $CO_2$ fluxes is then compared with the IAV of the biosphere flux models (VPRM and FLUXCOM) used as priors in the two inversion runs.

Seasonal NEE anomalies between 2006 and 2019 (ΔNEE, Fig. 8) are positive in the years 2007, 2010, 2016, 2018 and 2019

indicating that the mean uptake of $CO_2$ during the growing season (summer) was lower than average in these years. The magnitudes of anomalies during these years are comparable with findings of NEE anomalies reported by Rödenbeck et al. (2020), but estimated using different global inversion runs (2019 was not included in that study). Herein, we shed more light on 2019 NEE estimates which suggest even a weaker uptake of $CO_2$ in comparison with the summer of 2018. It is noticed that the posterior fluxes estimated using the biosphere model FLUXCOM exhibit the largest anomaly of NEE during the summer

of 2019. Despite slight differences in the amplitude of IAV, there is good agreement in the posterior fluxes of both the FLUXCOM and VPRM models. Such common agreement is inherited from the identical observations used in both the inversion runs, as demonstrated in case of the biosphere ensemble in section 3.1. Therefore, the IAV in this case is more likely to be attributed to climate anomalies, in particular during drought occurrence in the growing season.

The agreement between posterior fluxes using FLUXCOM and prior fluxes of VPRM in the spring season confirms two

important conjectures: 1) posterior IAV are largely derived by atmospheric data regardless of the biosphere model used, 2) the VPRM model can capture year-to-year variations during spring, reflecting its capability to represent dynamic biospheric activity during the growing season. It is clear that FLUXCOM exhibits remarkably weaker annual variations during spring and fall in comparison with the VPRM and the a-posteriori. In winter, VPRM model agrees well with FLUXCOM in the interannual variations, showing less IAV compared to the NEE estimates. We attribute this to the lower signal of temperature assimilated

in the biosphere models from the meteorological data, as well as less information of radiation reflectance obtained from the remote sensing data due to dominant cloudy scenes in winter, provided that the VPRM and FLUXCOM models use forcing data from meteorology and remote sensing.

To assess the temporal changes of NEE in response to such climate variations, we compare the seasonal anomalies of NEE (prior and posterior) to the anomalies of 2-m air temperature and Standardized Precipitation and Evapotranspiration Indices

(SPEI) (Beguería et al., 2014) during spring, summer, fall, and winter, as well as the annual mean (Fig. 9). Here we show estimated NEE integrated over the full domain. Monthly near-real-time data of SPEI (SPEI01) are obtained from https://spei.csic.es/map/maps.html at 1° spatial resolution and monthly 2-m air temperature accessed from https://psl.noaa.gov/data/gridded/index.html at 0.5 spatial resolution. The anomalies were normalized with the standard deviation of the interannual variations since 2006. In addition, Pearson correlation coefficients between posterior fluxes, prior

fluxes, temperature and SPEI for the full year and calendar seasons were calculated. Of note, due to the fact that the biosphere model VPRM utilizes temperature from meteorological fields and EVI data from the satellite sensor MODIS, it is anticipated to systematically correlate with temperature and SPEI. Consequently, we mostly devote our comparison to the posterior fluxes. The findings of standardized anomalies in Fig. 9a show that the decrease of $CO_2$ uptake in 2018 and 2019 summers (positive NEE anomalies) was exceptional and concurrent with a profound deficit of SWC (negative SPEI anomalies, dry conditions).

The reduction/very low SWC also coincided with an unprecedented rise of temperature (positive T anomalies, highest in 2018)





across Europe. Being an indicative factor of drought occurrences, SPEI links water availability in the surface including soil moisture (crucial limitation of GPP, especially in the temperate regions) and temperature to the precipitation and evapotranspiration rate. Hence, there is quite good agreement between posterior NEE and SPEI not only at spatial scales but also at temporal scales. The standard deviations of the interannual variations of posterior NEE, SPEI, and temperature over all

Europe in the annual mean through the 14 years were equal to 0.17 PgC yr$^{-1}$, 0.12, and 0.45 K, respectively. When relating the changes occurring during 2018 and 2019 to the context of the previous 12 years, the annual anomalies of SPEI found to decline to more than twice as much as the climatological deviation around -0.29 in 2018 and to around -0.08 in 2019. As a consequence, posterior NEE anomalies increased to 0.14 and 0.08 PgC yr$^{-1}$ above the climatological mean in 2018 and 2019, respectively. The excess of annually averaged temperature was predominant in 2018 and 2019, reaching around 0.40 and 0.47 °C above the

climatological mean, respectively. Despite the fact that the impact of the 2018 and 2019 drought on NEE is realized from the SPEI and temperature anomalies, there is a relatively moderate correlation between estimated NEE and SPEI and temperature at the annual scale over the full domain (Fig. 9b). However, the correlation coefficients largely vary between seasons. A high anticorrelation (-0.86) between estimated NEE and temperature is found to be consistent during spring. In contrast, anticorrelation drastically decreases and turns out to almost vanish in summer and fall. Nonetheless, the relatively moderate

correlation of SPEI with posterior NEE during summer is adequate to deduce the lack of SWC under dry conditions through the anticorrelation between SPEI and temperature. This implies, warm conditions accelerate the depletion of soil moisture content, in particular on the soil top layer that lacks water content in its deeper layers to compensate for the higher evaporation rate at the surface layer. This affects photosynthesis efficiency during the growing season by decreasing the gross primary productivity, but also increases the contribution of soil respiration, more pronounced in 2019.

During winter, water availability does not seem to be a limiting factor of NEE as we notice from low correlations between posterior NEE and SPEI. Instead, temperature negatively correlates with posterior NEE indicating that the increase of temperature coincides with enhanced uptake of $CO_2$ where photosynthesis can occur, e.g., in evergreen areas. But cold years have more and longer snow cover which decreases photosynthesis, whereas soil heterotrophic respiration contributes more to $CO_2$ release since soil temperature is expected to be larger than air temperature owing to snow cover insulation. Figure 10

illustrates the seasonal contribution of NEE to IAV, which is dominated by summer and spring variability in comparison with winter. We note that the posterior fluxes are in agreement with the biosphere model VPRM in summer and fall, while the variability of posterior fluxes is larger during winter and smaller during spring compared to prior fluxes; the opposite holds true for the prior fluxes. Temperature is, however, shown to largely vary during winter, while SPEI contribution does not show a significant variability between seasons.

**Spatial differences of NEE estimates in 2018 and 2019**

Using identical observations for the last 5 years, S2 inversion demonstrates the differences between NEE estimates in 2018 and 2019 as noticed from Fig. 11. Results emphasise the aftermath of drought episodes, showing a smaller uptake of $CO_2$ in France, Germany, and Northern Europe during spring of 2018 (March-April-May), while during summer of 2019 (June-July-August) the estimates of NEE, to some extent, suggest a higher $CO_2$ release, in particular in the United Kingdom, France,

Germany, and Southern Europe. Obviously, during winter time (December-January-February) the differences are infinitesimally small throughout the full domain, while the annual mean fluxes indicate much smaller uptake in 2018 compared to 2019. This confirms a longer impact of the drought lasting from the early growing season during spring until the end of summer. To explain the changes of spatial distribution of NEE alongside SPEI and temperature in 2019, anomalies of NEE were estimated using S1 inversion with respect to 2006-2018 anomalies. Figure 12 indicates the coincidence of the large

release of $CO_2$ during summer time in Central Europe and in the United Kingdom (if we ignore the anomalies over Spain due to absence of observations) with the positive anomalies of temperature and the negative anomalies of SPEI over those regions. The prior fluxes, to a lesser extent, show the impact of temperature and SWC on NEE during the growing season. However, in the United Kingdom positive NEE anomalies can only be detected from the posterior fluxes. The positive anomalies of temperature during winter show a slight impact on NEE (positive anomalies). This can be interpreted as an increase of soil

respiration.

The annual budgets of NEE in 2019 and 2018 are summarized in Fig. 13 for six subregions estimated using B0 and S2 inversions. The choice to use B0 inversion is to estimate annual flux budgets of $CO_2$ through assimilating as many atmospheric observations as possible to strengthen the observational constraint in the spatial and temporal aspects. S2 inversion is specifically used to keep identical observations in 2018 and 2019 for the purpose of assessing the NEE differences between

the two years. The annually aggregated fluxes estimated from B0 inversion over the full domain yield -0.28 and -0.22 PgC in 2019 and 2018, respectively. The performance of the inversion reflected in the a-posteriori fluxes is associated with an uncertainty reduction of 85.5 to 88% with respect to assumed prior error for 2019 and 2018, respectively. Likewise, the underlying European regions indicate uncertainty reduction with different magnitudes based on atmospheric data availability. The relatively observational weak constraint in 2019 has thus resulted in a small increment of posterior uncertainty in regions

that have a fewer number of stations. For instance, the uncertainty in Southern Europe amplified from 0.018 PgC in 2018 to 0.027 PgC in 2019, coincided with an increase of the net source of $CO_2$ fluxes from 0.016 PgC in 2018 to 0.069 PgC in 2019. Despite the data coverage difference in both years, Southern Europe still shows a larger annual flux of 0.04 PgC in 2019 estimated using S2 inversion in comparison with 0.02 PgC in 2018.

These results, again, highlight the sensitivity of the inversion to data coverage, but also the stronger impact of warmer summers

on NEE, where S2 estimates suggest larger flux budgets in 2019 compared to 2018 over Western and Southern Europe. Overall, B0 and S2 results suggest a suppression of GPP, predominantly in Central and Northern Europe.

## 4 Discussion and conclusions

### 4.1 Sensitivity of posterior fluxes to input data

The smaller spread in the a-posteriori fluxes found through the ensembles of inversions is evident over All Europe reflecting

the good performance of the inversion system. In the biosphere ensemble, flux estimates are not very sensitive to a-priori terrestrial ecosystem fluxes. We deduce this from the small spread over the a-posteriori fluxes (Fig. 3), occurring despite major





differences in a-priori fluxes. Likewise, different ocean flux models have the smallest effect on estimating NEE, in particular inland, where ocean-land exchange is dissipated. However, the spread in the station set ensemble is strongest, at 0.11 PgC yr$^{-1}$ for the annually aggregated fluxes over the full domain. This points out a higher sensitivity of the inversion to the number of

stations in comparison with 0.06 and 0.02 PgC yr$^{-1}$ spreads in the biosphere and ocean ensembles, respectively. This effect is most pronounced over Central Europe, where measurements of $CO_2$-dry-mole-fractions are available from a large number of stations, and thus a contrasting number of sites manifests amongst station subsets, given that the station subsets were not selected based on geographical locations but on the long record of data coverage. Further, this finding corroborates the dominant influence of observational constraint on NEE estimates in the biosphere and ocean ensembles seen through

interannual variations. Such influence is, however, subject to the availability of the atmospheric data which can otherwise be altered by the prior constraint to maintain the posterior estimates according to Bayes' approach. For example, the relative spread over biosphere ensembles in Northern Europe, having less dense coverage, increases to 38.4%, compared to 23.9% in Central Europe. Conversely, in the stationset ensemble, the spread calculated for Northern Europe was found to be smaller than that in Central Europe (51.8% versus 71.7%), reflecting the biogenic flux constraints in this case, given the smaller

differences of observations among the station subsets in this ensemble. The ocean ensemble spread remains at a minimum percentage in both regions, albeit it elevates to 9.6% in Northern Europe in comparison with only 4.3% in Central Europe. Although the impact of ocean fluxes on NEE estimates can be negligible over the full domain and inlands, the results point to a relatively higher influence in the coastal regions.

Our results denote a comparable reduction of posterior uncertainty and the spread relative to their a-priori (Fig. 5). Noteworthy,

the indirect effect of statistical posterior uncertainty on the corresponding spread over the ensembles of inversions emerges from the common dependency on observational data, which predominantly appear in the well-constrained areas in Germany, France, Benelux, and the U.K. Over such regions, the posterior uncertainty and spreads are greatly reduced and the inversions tend to converge regardless of which prior flux model is used. It is essential to consider the prior error assumption, as well as the prior error structure in the spatial and temporal aspects. This will determine to which extent the posterior fluxes are

dependent on the prior biogenic fluxes, specifically in the regional inversions where the degrees of freedom can drastically increase following the finer spatial and temporal resolution of biosphere flux models and atmospheric transport models.

## 4.2    Response of NEE estimates to climate variation

The linkage between NEE and climate variation has been examined via SPEI and temperature as proxy data of climate variation. The anomalies of SPEI and temperature are analysed along with NEE anomalies during the recent years 2018 and

2019 in the context of the period 2006-2017. The recent drought events decreased the efficiency of GPP, in particular during spring and summer, where soil moisture markedly declined during the summer of 2018 and 2019 accompanied with an exceptional rise of temperature (Ma et al., 2020). But GPP during 2019 spring showed a higher efficiency (larger uptake of $CO_2$) than the spring of 2018, benefiting from the increment of temperature, SWC, and light availability. The finding is consistent with a study on seasonal NEE over North America implemented by Hu et al. (2019) and seems to hold for Northern



regions where temperature is substantially considered as a limiting factor to NEE. Additionally, our results showed that Central Europe experienced higher sources of $CO_2$ during 2019, which can be impacted by an extended legacy of the drought of 2018, where forests were profoundly stressed and thus their growth was negatively impacted. MacKay et al. (2012) found about 17% reduction of drought plot growth relative to a reference plot and showed that growth reduction in the forests across Europe exceeded this value under drought conditions depending on tree species. Furthermore, the ecosystem respiration response to

the temperature increment may contribute to such a positive anomaly, given that temperature anomalies during 2019 over Central and Southern Europe were unprecedented in line with the 2018 anomaly (Hari et al., 2020). In agreement with Rödenbeck et al. (2020), we found that summer NEE anomalies were in agreement with the anomalies of temperature and SPEI, occurring in different summers including 2018.

In terms of estimated winter fluxes, the medium anticorrelation between temperature and NEE (also shown by the anomalies

in Fig. 9) implies that an increase of $CO_2$ release occurs in cold winters when thicker and lasting snow cover prevail. Snow insulation prevents the soil temperatures from decreasing as strongly as air temperature. In comparatively warmer soils respiratory carbon dioxide emissions by the soil biota are sustained also during winter. Being in agreement with our results, Monson et al. (2006) found that soil temperature during winter increases nonlinearly with the depth of snowpack resulting in enhancement of soil respiration. Even though the study was carried out at site level in the Northern Hemisphere in the U.S, the

agreement with our findings refers to a similar impact at a widespread scale.

Besides the previous explanation of T-NEE anticorrelation in winter, one should take into account a hypothesis of the vertical mixing height being systematically biased in the atmospheric transport models. In this case, a misrepresentation of mixing height during colder winters might occur, as was demonstrated in a study conducted by Gerbig et al. (2008) devoted to characterizing the uncertainty of atmospheric transport models resulting from vertical mixing changes during day and night

time. Given that, the shallow boundary layer developed during night holds similar characteristics during cold winters. A separate study is required to investigate such an impact which will lead to improving vertical mixing in the atmospheric transport models, and thus reducing the uncertainty in atmospheric tracer inversions. Apart from that, the contribution of winter fluxes to the interannual variations is lower than that resulting from summer fluxes, denoting a lesser impact.

Overall, the NEE response to SWC and temperature varies depending on the temporal and spatial aspects of the region of

interest, and is connected to the hydrological cycle and physical dynamics of soils and the ambient atmosphere.

In this paper, the NEE flux budgets of 2018 and 2019 are estimated in a pre-operational method to keep track of the changes of net terrestrial fluxes of $CO_2$. Results still suggest the domain of Europe as a net sink of $CO_2$, albeit in 2018 and 2019 $CO_2$ uptake decreases to $-0.22 \pm 0.05$ and $-0.28 \pm 0.06$ PgC yr$^{-1}$, respectively, due to drought occurrences compared to a multi-year average uptake of $-0.36 \pm 0.07$ PgC yr$^{-1}$ (2006-2019). In contrast to the a-posteriori, the prior fluxes obtained from VPRM are

by far overestimating $CO_2$ uptake, especially in the growing season, resulting in $-1.47 \pm 0.43$ and $-1.37 \pm 0.43$ PgC yr$^{-1}$ in 2018 and 2019, respectively. The posterior fluxes constrained in the regions or individual countries whose observational data are denser, such as France, Germany, and the U.K, are more reliable than those that have sparse observation network (e.g., Poland or Spain) or do not have any monitoring sites at all (e.g., Turkey). This implies that NEE in some countries within the domain



of Europe is constrained either by weak observational signal from the neighbouring regions through the spatial correlation

length of prior error, or mostly dominated by the prior flux constraint. This kind of flux constraints skews the total aggregated fluxes at the expense of well-constrained regions and is expected to amplify the posterior uncertainty. For example, when comparing 2019 NEE budgets in France and Turkey, the a-posteriori fluxes are summed up to -0.032 ± 0.008 and 0.045 ± 0.020 PgC yr$^{-1}$, respectively. Despite the fact that they have different areas, the uncertainty in the latter is about 45%, notably greater than that realized in the first, up to 25%. These results emphasise the importance of using a wider coverage of $CO_2$

observations in the regional inversions to better estimate the continental flux budgets, and thus understanding the biogenic flux changes amid climate variation.

**Code and Data availability**

NEE estimates and the respective prior fluxes, as well as codes used in this study can be made available upon request.

Atmospheric dry mole fraction measurements of $CO_2$ are available from the ICOS Carbon Portal and can be accessed from http://doi.org/10.18160/GZ5S-4GPR. SPEI01 can be downloaded from the near-real-time data through https://spei.csic.es/map/maps.html.

**Competing interests**

Some authors are members of the editorial board of journal Atmospheric Chemistry and Physics. The peer-review process was guided by an independent editor, and the authors have also no other competing interests to declare

**Acknowledgements**

The Authors acknowledge the compactional support of Deutsches Klimarechenzentrum (DKRZ) where the CSR inversion

system is implemented; the providers of atmospheric dry mole fraction measurements of $CO_2$ represented by ICOS and NOAA site networks available across Europe. We thank Wouter Peters for providing NEE fluxes from SiBCASA model.

**Financial support**

This research has been supported by the Horizon 2020 (VERIFY, grant no. 776810)

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





**Figure 1: Stations network distribution over Europe. Different graphical symbols denote the type of station classifications; coloured regions indicate Central Europe (green), Northern Europe (blue), Western Europe (purple), Southern Europe (orange), Eastern Europe (yellow), and South-eastern Europe (light red).**




**Figure 2: Dataset density measured from 2006 until 2019 over Europe. Yellow to red colour scale denotes monthly-averaged dry mole fractions of CO₂. Symbols on the right-hand axis: C - core site, R - recent site.**




**Figure 3: NEE fluxes estimated using B0, B1, B2, S1, and O1 inversions for the 2006-2018 period over the full domain of Europe (top), Central Europe (middle), and Northern Europe (bottom). Posterior fluxes are plotted with solid lines and their a-priori in the dotted lines. Priors and posteriors of the biogenic ensemble are distinguished by identical colours for each modelled scenario. Light red shadowing denotes the statistical uncertainty and error bars indicate the spread among the biosphere inversions' ensemble.**






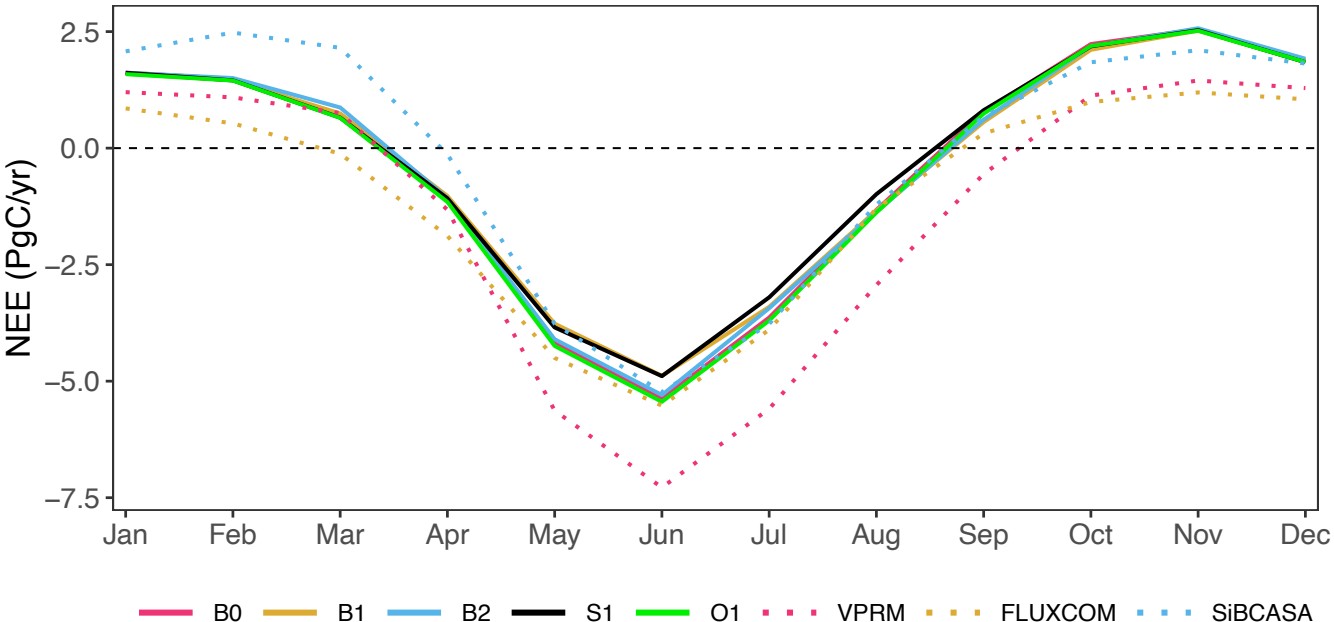

**Figure 4: Seasonal cycle of NEE calculated as the average of monthly fluxes over 13-year estimated using the ensembles of inversions (solid lines) B0, B1, B2, S1, O1, as well as the biogenic prior fluxes (dotted lines) obtained from VPRM, FLUXCOM, and SiBCASA.**

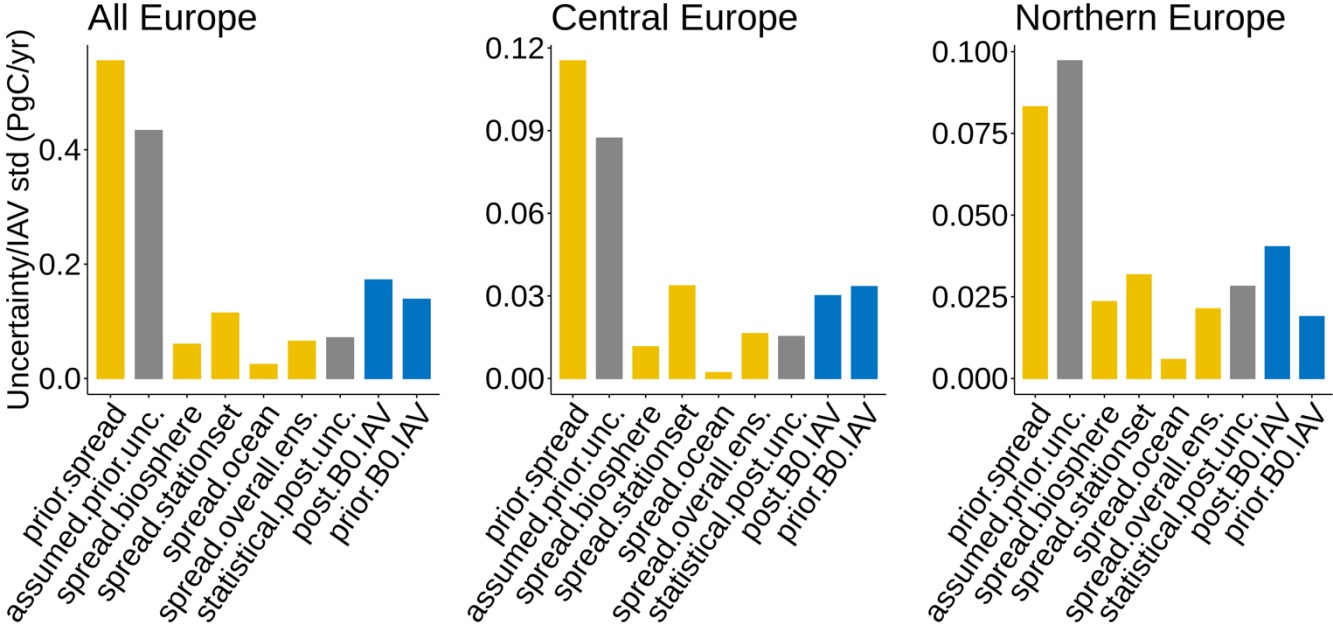

**Figure 5: Spread uncertainties calculated from 3 inversion ensembles of biosphere, ocean, and station set (yellow bars). Grey bars refer to the statistical uncertainties, and blue bars denote the standard deviations of IAV.**







**Figure 6: Spatial spread of biosphere, prior, station set, and ocean ensembles. Standard deviation (sd) on the legend is normalized over maximum spread 1.91×10⁻⁴ in units of PgC yr⁻¹ per grid pixel. Stations used in the station set ensemble are referred to by circles (B0), dots (S1) and plus symbol (S2).**





**Figure 7: Posterior, prior, and innovation of fluxes (posterior - prior) averaged over the 2006-2018 period calculated from the biosphere ensemble of inversions (B0, B1 and B2). Green circles refer to observing sites.**





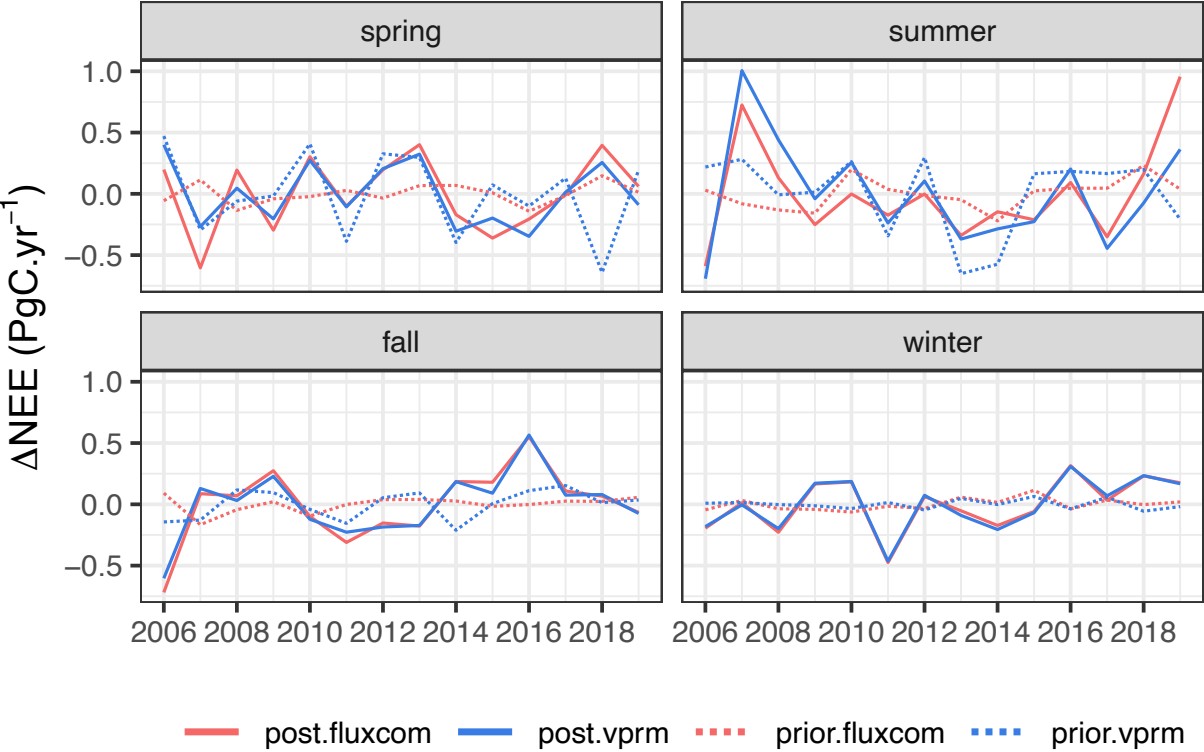


**Figure 8: Anomalies of NEE fluxes during spring, summer, fall and winter estimated from two inversion runs differing in biosphere models (FLUXCOM in red colour and VPRM in blue colour) using the atmospheric data of core sites. Solid lines indicate the a-posteriori while dashed lines refer to the corresponding a-priori (biosphere models).**





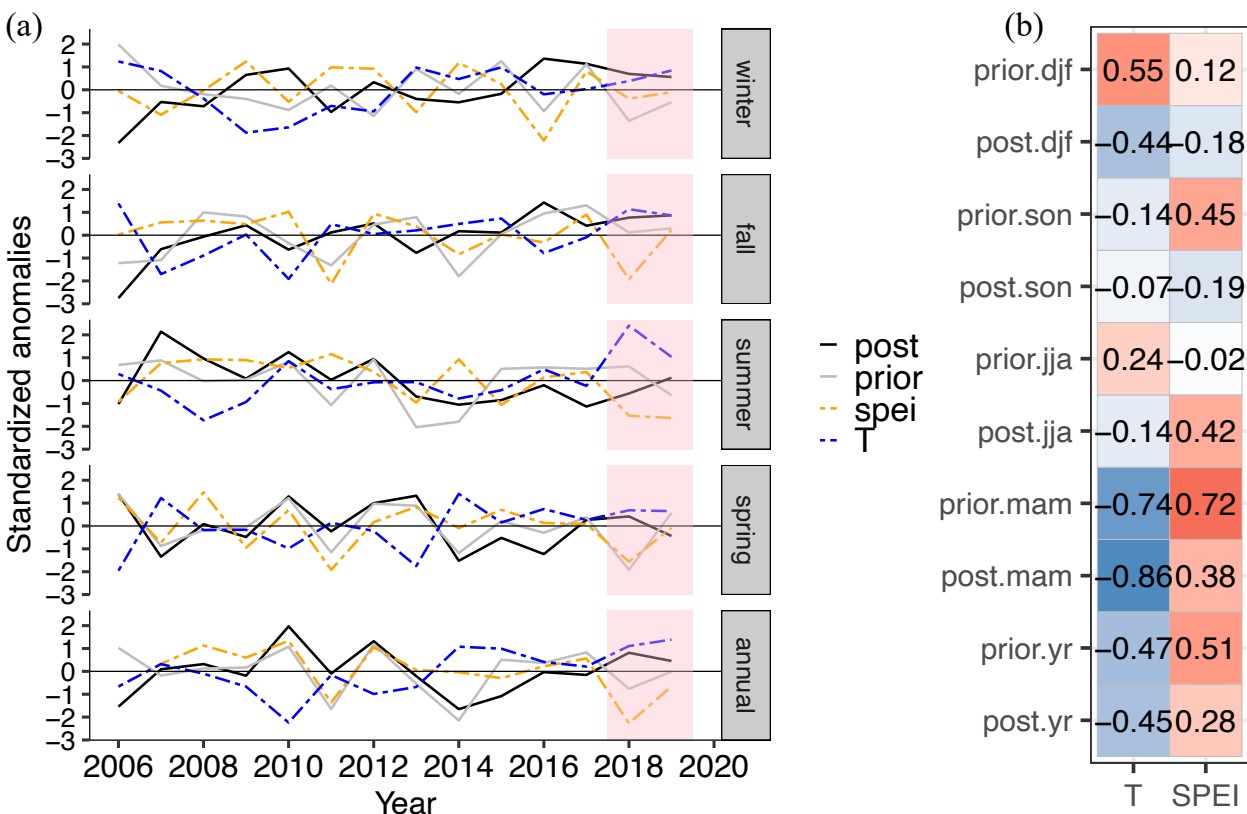

**Figure 9: Left panel (a) shows the anomalies of posterior NEE (post), prior NEE (prior), SPEI (spei) and 2-m air temperature (T) standardized relative to the standard deviation of climatological variations at annual and seasonal scales since 2006 over the full domain of Europe. Units of NEE and temperature are in PgC yr⁻¹ and K, respectively, while SPEI is unitless. Right panel (b) refers to the correlation coefficients of posterior and prior NEE on y-axis with SPEI and air temperature on x-axis calculated in springs (mam), summers (jja), fall (son), and winters (djf), and at annual scales (yr) over the full domain of Europe. Note: seasonal T and** 595 **SPEI correspond to NEE seasons mentioned on y-axis.**





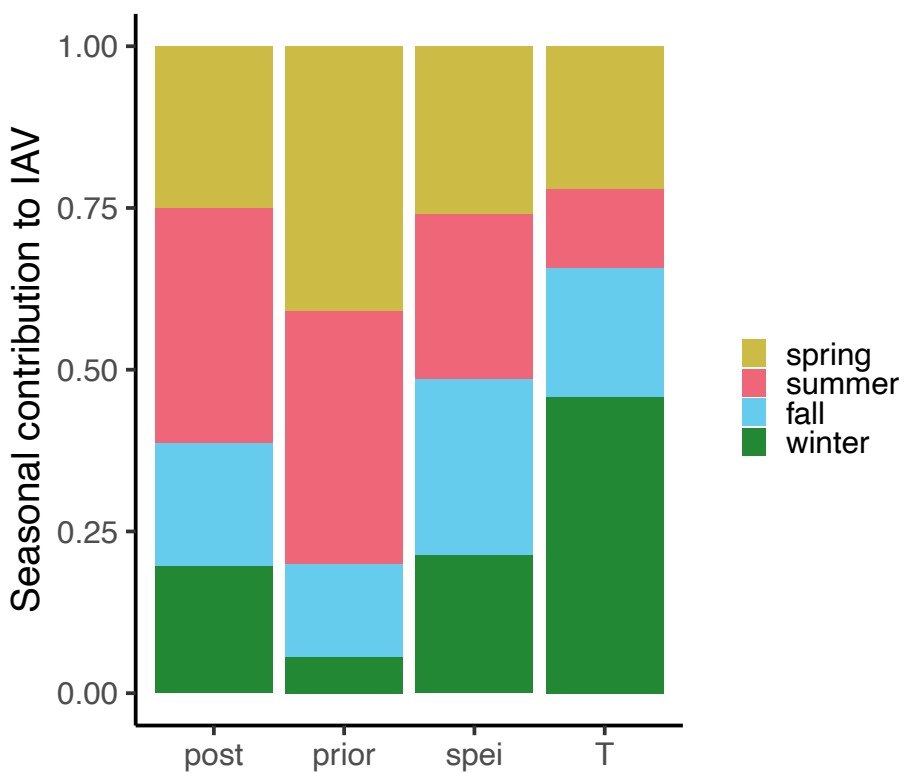

**Figure 10: Seasonal contribution to IAV calculated relative to 14 years for posterior NEE fluxes (post), VPRM NEE fluxes (prior), SPEI, and T during the four seasons over the full domain.**

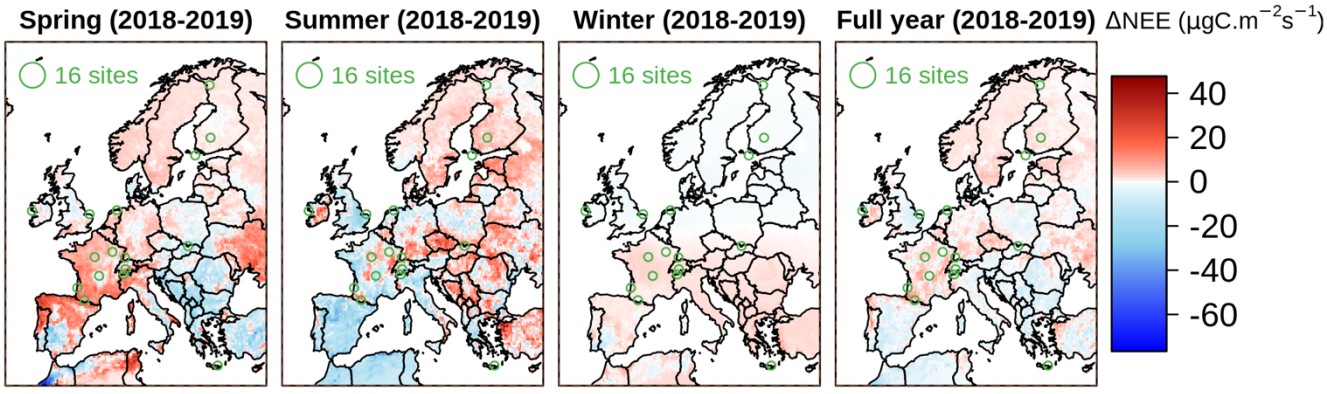

**Figure 11: Differences of NEE estimates for 2018 – 2019 in seasonal and annual mean calculated from S2 inversion setup.**







**Figure 12: 2019 anomalies of prior fluxes (first row), posterior NEE estimated from S1 inversion (second row), 2-meter air temperature (third row), and SPEI (fourth row) relative to 2006-2019 over Europe. Columns denote mean estimates of spring, summer, fall, winter and annual NEE estimates, from left to right.**



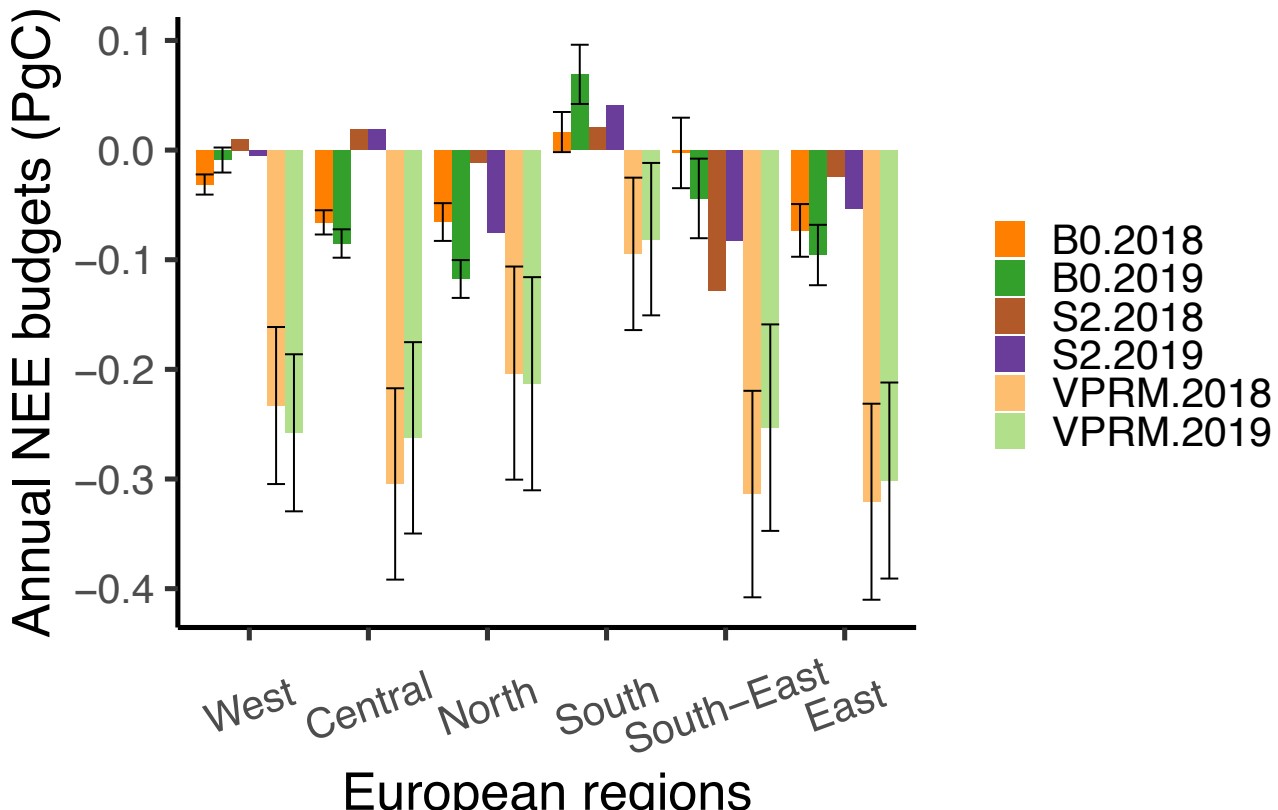

**Figure 13: Posterior NEE flux budgets over 6 European regions in 2018 and 2019 using B0 and S2 inversions (dark colours)**
**compared to their priors from the VPRM model (light colours). Uncertainties associated with B0 and VPRM fluxes are referred to in the error bars.**

**Table 1: Representation error of station locations**

| Classification | Mountain | Tower | Ocean | Continental | Urban |
|---|---|---|---|---|---|
| **Code** | M | T | S | C | UP |
| **Error (μmol mol⁻¹)** | 1.5 | 1.5 | 1.5 | 2.5 | 4 |

**Table 2: Set-ups of the inversions**

| Inv. code | Biosphere | Ocean | Station-set | Time period |
|---|---|---|---|---|
| **B0 (base)** | VPRM | Mikaloff | all | 2006-2018 |
| **B1** | FLUXCOM | Mikaloff | all | 2006-2018 |
| **B2** | SiBCASA | Mikaloff | all | 2006-2018 |
| **O1** | VPRM | Carboscope | all | 2006-2018 |
| **S1** | VPRM | Mikaloff | core sites | 2006-2019 |





| S2 | VPRM | Mikaloff | recent sites | 2015-2019 |


**Table 3: Reduction of the biosphere ensemble NEE spread over the full domain, Central and Northern Europe**

| Region | Prior spread (PgC/yr) | Posterior spread (PgC/yr) | Spread Reduction (%) |
|---|---|---|---|
| **All Europe** | 0.666 | 0.032 | 95.1 |
| **Central Europe** | 0.137 | 0.005 | 96.0 |
| **Northern Europe** | 0.098 | 0.024 | 74.8 |