# Peer review of "NEE estimates 2006-2019 over Europe from a pre-operational ensemble-inversion system"

_Atmospheric Chemistry and Physics, 2021_

## Referee Comment (RC2)

Review of Munassar et al. 2021 submitted to Atmospheric Chemistry and Physics

**General**

The study presents inverse modelling estimates of the net ecosystem exchange across Europe for the period 2006 to 2019 with a special focus on the exceptional years 2018/2019 and the analysis of influence of environmental drivers (temperature, moisture availability) on NEE. The applied methods have been established in previous work and are generally sound. However, a few general questions concerning the method should be addressed (see comments below). The manuscript is well organized and generally well written. Nevertheless, in some cases the description of results could be more precise. Overall, I recommend the manuscript for publication after a series of rather minor issues have been addressed.

**General comments**

Title: In the title and elsewhere the inversion system is labeled 'pre-operational'. However, there is no discussion whatsoever, why this term is used. Furthermore, if the current system is the pre-operational system, what do the authors envision as the operation system? Either this label has to be discussed in more detail or it should be removed from the text and title.

Anthropogenic emissions: Anthropogenic CO2 emissions are assumed to be well-known for the current study (L51) and are not updated as part of the inversion. How sure can we be about this? While the sensitivity of the inversion towards prior biospheric fluxes and prescribed ocean fluxes is investigated, no such analysis is done for the anthropogenic fluxes. Some kind of discussion of this assumption should be included in the manuscript. What is the potential uncertainty added by this assumption. This, especially in the light of inter-annual variability and the timing of anthropogenic emissions that exhibit larger uncertainties than annual national totals.

**Specific comments**

L44-46: Consider adding references to this statement.

L53 and elsewhere: The terms 'error' and 'uncertainty' seem to be used interchangeable throughout the manuscript. Consider using 'uncertainties' throughout and distinguish between random and non-random uncertainties, if that was the intention when using error (systematic) and uncertainties (random).

L83/84: Since 2006 IFS HRES has undergone several configuration changes, some of them affecting horizontal and vertical resolution. Please mention at which resolution the input data was available (spatial and temporal) for STILT and if and when configurations changed.

L85: The number of released particles seems very small. Over which time interval are these particles released? Other inverse modelling studies using Lagrangian transport models employ much larger particle numbers (e.g. Lauvaux et al. 2016) even though their domains and transport times are smaller. If only 100 particles are released per hour of measurements but residence times are evaluated in a grid of approximately (160 x 160 = 25'600) cells, it seems unlikely to get a statistically robust estimate of residence times.

L110: Depending on the height of the mountain, I would rather say that these sites experience free tropospheric conditions than residual layer conditions.

L145/146: What is this choice based on? Which values did these length scales take? Are these decay distances constant over the whole domain?

L150: Were the CarboScope ocean fluxes updated since the 2013 publication or were these also climatologies?

L152: What is the benefit of using EDGAR 4.3 over the much more recent EDGAR 6.0, which provides more temporally resolved emissions up to 2018? Since these emissions are taken as 'truth', I wonder if a sensitivity inversion with alternative anthropogenic emissions should have been conducted. Would it be possible to make an estimate of how large an uncertainty may be added by the prescribed anthropogenic emissions?

L165: The sample standard deviation when using only two or three samples is not a very robust estimator of the true standard deviation and is generally biased low. This should be stated as a warning when comparing these ensemble spreads with 'true' standard deviations as used in prior uncertainty.

L167/168: In Bayesian inversions, it is usually possible to calculate the posterior covariance directly. It sounds as if this is not the case here. The posterior uncertainty will also depend on the prescribed prior

uncertainty. Here, it seems that the prior uncertainty is not considered but only the data-mismatch uncertainty is used. Please clarify.

Section 3.1: Results in this section only span the period 2006-2018 and exclude 2019. Although, this is correctly listed in Table 2, this limitation is not mentioned in the text and the title of the manuscript suggests that results would include 2019. Please clarify in more detail that different periods are used and why.

Section 3.1: Results in this section are only presented for Central, Northern and all of Europe. It would be interesting to see the results for the other regions as well. Could similar figures as Fig 3 and 5 also be provided for the other regions as part of a supplement? Especially since there are some references in the text to other regions (e.g. L191).

Table 3: Provide results for all of Europe. How is it possible that spread reduction is 95.1 % across all of Europe if Central Europe has a spread reduction of 96% and all other regions will have considerable less reduction? Without seeing the results for all European regions, the discussion remains unsatisfactory.

L216: This is already discussed in L195, where a reference to table 3 (with the same numbers) is given. Please consolidate.

Figure 7, L242-247: How is it possible that the innovation over south-eastern Europe is almost as large as in Central Europe even though there are no observations in this area (B0, B1)? Should the posterior in that case not stay very close to the prior?

L249: This sounds as if a different inversion system as compared to the previous section was used. Please clarify.

L250: It is the same starting time as in the previous section. Only the additional year 2019 seems different.

L259: 'Seasonal NEE': Did you mean summer? Otherwise, the described years with positive anomalies don't make sense.

L274/275: The biospheric signal is generally weak in winter (as predicted by the biosphere parameterisations). The IAV seen in the posterior may also be attributable to IAV in fossil fuel emissions that are not well represented in the used inventory. Please comment.

L283: Why was this temperature dataset selected over ERA-5, which was used to drive the FLUXCOM estimates?

L288/289: I do not see this at all. The summer anomalies in earlier years (2007, 2010, 2012) were much more dramatic. Even if we discard these because of the poorer data coverage (but why show them then in the figure?), summer 2018 does not seem too exceptional. Wouldn't it make more sense to discuss the growing season as a whole instead of summer and spring separately? Spring 2018 and 2019 look more exceptional to me than the summer months. Or limit the discussion to the whole year as is done below.

L312: Could this also be driven by a generally earlier start of the growing season (not just in evergreens) already towards the end of February? Especially in southern Europe many crops start developing around that time already. Finally, this may once again be a misattribution of anthropogenic emissions. Warmer winters mean less anthropogenic emissions, if these are not considered correctly or fully the "missing" CO2 may be attributed to biospheric uptake!

L321: 'identical observations'. Do you mean an identical set of observation sites? Otherwise, this could be misunderstood in the sense that identical observations were repeatedly used each year.

L321: Why are none of the German ICOS sites used here? Were they not available in 2018 and 2019? Why restrict to only 16 sites when trying to analyse the spatial differences?

L329-331: This is not correct. Over the UK we can see basically no temperature anomaly in summer 2019, a positive SPEI anomaly but still largely increased posterior NEE. Please be more precise in the description.

L400: Most of western and central Europe do not experience any lasting snow cover anymore! Periods with snow cover are mostly limited to a few days. Colder winters also don't necessarily mean more snow as cold periods in central Europe are usually connected to easterly advection in high pressure systems with little precipitation. Overall, this 'theory' would need to be evaluated with additional datasets (snow cover, soil temperatures, etc.).

Figure 11: Why exclude fall here?

Definition of winter season: Which months are incorporated into the winter estimate of a specific year (X)? Jan X, Feb X and Dec X, or Dec X, Jan X+1, Feb X+1? Please clarify. If the first definition is used then there is no connection in the climatological sense and the interpretation may be more difficult.

**Technical comments**

L38: NEE was only defined in abstract. Please redefine in main text.

L103: Remove line. Seems to be a mistake.

---

## Author Comment (AC1)

**Final authors' response (AC) to the referee comments (RC1) on acp-2021-873**

*We are very thankful to the reviewer for the constructive comments and for considering the manuscript for publication in ACP after minor revisions, which we have done according to the suggestions of reviewer. In the following, we address the reviewer comments under the respective sections — i.e., General comments, Detailed comments, and Technical corrections.*

*Note: The reviewer comments (RC1) are referred to in "Arial" font type throughout the texts, and the authors' responses are referred to as "Italic Arial" with indented lines.*

**Anonymous Referee #1 (RC1)**

Referee comment on "NEE estimates 2006–2019 over Europe from a pre-operational ensemble-inversion system" by Saqr Munassar et al., Atmos. Chem. Phys. Discuss., https://doi.org/10.5194/acp-2021-873-RC1, 2021

**1) General comments**

Authors present an analysis of the European terrestrial carbon cycle variability in 2006-2019 made with the pre-operational inverse modelling framework "CarboScope Regional". The $CO_2$ flux estimates are shown to be largely independent from the prior fluxes in the area of dense observations. The results confirm dominance of the observational constraint on fluxes and the importance of climate controls on the interannual flux variability. Authors find the inverse model predicts statistically significant positive $CO_2$ flux anomalies in 2018-2019 related to hot and dry climate in those anomalous years. The paper is well written and can be considered for publication after minor revisions.

> *Thank you for considering the paper to be published in the ACP journal. We have revised our manuscript accordingly as detailed below.*

**2) Detailed comments**

L75-80 Although some of the information can be found in references, to improve readability it is useful to give few more details about the CSR such as optimization scheme and temporal resolution of flux corrections.

> *We agree that adding more information about the optimization scheme is useful, so a complementary description is added in the methods in the revised manuscript (L85-101).*

L104 Need to give detail – where station types come from.

> *The station types are categorized with different classes according to the ability for the regional transport model to reasonably simulate the atmospheric concentration, given the variable complexity to represent the local circulation, over each station as explained in Rödenbeck, (2005). We have added this information in the revised manuscript (L130-132).*

**Final authors' response (AC) to the referee comments (RC1) on acp-2021-873**

L248-L300 The correlation of posterior fluxes with climate indices has been reported in detail. To enhance the validation of interannual flux variability, can authors add comparison with interannually varying regional flux estimates by independent process-based models, and possibly, top-down?

> *We have added an additional comparison on interannual variability of flux estimates in the supplementary materials (subtitled as "validation test on posterior IAV", also Fig. S3) using an independent top-down model (LUMIA inversion) which serves as a validation of IAV in our results. The validation showed good agreement in the IAV between both estimates despite different data inputs and inversion setups between both inversion systems.*

L407 Need a reference here on systematic bias in transport models.

> *Reference has been added in L488.*

**3) Technical corrections**

L18 Phrase 'We further investigate the unprecedented increase of temperature …' is somewhat incomplete, better write that one investigates the impact of 'unprecedented increase ..' on the carbon cycle.

> *This has been rephrased accordingly (L18-19).*

L103 'South-eastern Europe (light red).' Line out of place.

> *This line is removed (L130).*

L265 'fluxes of both' can be replaced with 'fluxes estimated with both'

> *We changed it in the revised version, L326.*

L405 'widespread scale' can be reduced to 'wide scale'

> *It is changed based on the suggestion, L486.*

L424-425 The phrase 'spatial correlation length of prior error' can be reformulated, it would be more accurate to avoid using 'prior' as this spatial correlation is applied to posterior flux corrections.

> *The spatial correlation actually belongs to the prior error, together with the temporal error correlation it forms the assumed error structure associated with the prior uncertainty (L506-507).*

L460 Paper number in Chevallier 2012b is missing (Global Biogeochem. Cycles, 26, GB1021, doi:10.1029/2010GB003974)

> *We corrected the reference information accordingly, L550.*

---

## Author Comment (AC2)

**Final authors' response (AC) to the referee comments (RC2) on acp-2021-873**

*We are very thankful to the reviewer for the constructive comments and for considering the manuscript for publication in ACP after minor revisions, which we have done according to the suggestions of reviewer. In the following, we address the reviewer comments under the respective sections — i.e., General, General comments, Specific comments, and Technical comments.*

*Note: The reviewer comments (RC2) are referred to in "Arial" font type throughout the texts, and the authors' responses are referred to as "Italic Arial" with indented lines.*

**Anonymous Referee #2 (RC2)**

Referee comment on "NEE estimates 2006–2019 over Europe from a pre-operational ensemble-inversion system" by Saqr Munassar et al., Atmos. Chem. Phys. Discuss., https://doi.org/10.5194/acp-2021-873-RC2, 2022
Review of Munassar et al. 2021 submitted to Atmospheric Chemistry and Physics

**1) General**

The study presents inverse modelling estimates of the net ecosystem exchange across Europe for the period 2006 to 2019 with a special focus on the exceptional years 2018/2019 and the analysis of influence of environmental drivers (temperature, moisture availability) on NEE. The applied methods have been established in previous work and are generally sound. However, a few general questions concerning the method should be addressed (see comments below). The manuscript is well organized and generally well written. Nevertheless, in some cases the description of results could be more precise. Overall, I recommend the manuscript for publication after a series of rather minor issues have been addressed.

> *Thank you for considering the paper to be published in the ACP journal. We have revised our manuscript accordingly as detailed below.*

**2) General comments**

Title: In the title and elsewhere the inversion system is labelled 'pre-operational'. However, there is no discussion whatsoever, why this term is used. Furthermore, if the current system is the pre-operational system, what do the authors envision as the operation system? Either this label has to be discussed in more detail or it should be removed from the text and title.

> *Pre-operational means that the system is used to provide annual updates of estimated fluxes. It is not an operational system as the system is under development from year to year.*
> *We have clarified that in the revised manuscript in L307-310.*

Anthropogenic emissions: Anthropogenic CO2 emissions are assumed to be well-known for the current study (L51) and are not updated as part of the inversion. How sure can we be about this? While the sensitivity of the inversion towards prior biospheric fluxes and prescribed ocean fluxes is investigated, no such analysis is done

**Final authors' response (AC) to the referee comments (RC2) on acp-2021-873**

for the anthropogenic fluxes. Some kind of discussion of this assumption should be included in the manuscript. What is the potential uncertainty added by this assumption? This, especially in the light of interannual variability and the timing of anthropogenic emissions that exhibit larger uncertainties than annual national totals.

*Evaluating emission uncertainty is still challenging as the truth of spatio-temporal emissions cannot not be easily reported. Nevertheless, it has been the standard approach in CO2 inversions to solve for the highly uncertain biospheric exchange fluxes and to assume that the anthropogenic emissions are well known (Rödenbeck et al., 2020; Peylin et al., 2013; Chevalier et al., 2012, Monteil et al., 2020).*

*A simple estimate of the uncertainty associated with anthropogenic emissions can be made when comparing the recently updated fossil fuel emissions over EU27+UK in 2014 as reported in Petrescu et al. (2021) obtained from eight data sources: BP, EIA, CEDS, EDGAR, GCP, IEA, CDIAC, and NGHGI (UNFCCC, 2019), the spread between the annual total emissions is about 0.038 PgC (with a mean of 0.974 PgC). The prior and posterior uncertainty of NEE amounts to 0.490 and 0.037 PgC per year, respectively. Prior NEE uncertainty is by far dominating in the inversion in comparison with emission uncertainty, which is within the same order of magnitude of posterior NEE uncertainty. This implies, the uncertainty in emissions would be about 4% whatever emission products we prescribe in the inversion among those abovementioned data sources. As a result, prescribing fossil fuel emissions in the inversion and solving for NEE is appropriate. However, when interpreting posterior biosphere-atmosphere exchange fluxes one has to take into account that part of the fluxes and their variability might be compensating for errors in anthropogenic emissions. Future releases of the pre-operational system will include different anthropogenic emission estimates and assess the resulting uncertainty in a separate study with more detail, making use of the recent emission products estimated from - e.g., TNO and EDGAR_v6.*

*We have discussed this assumption in the revised manuscript (L193-200).*

**3) Specific comments**

L44-46: Consider adding references to this statement.

*References added in the revised manuscript (L48).*

L53 and elsewhere: The terms 'error' and 'uncertainty' seem to be used interchangeable throughout the manuscript. Consider using 'uncertainties' throughout and distinguish between random and non-random uncertainties, if that was the intention when using error (systematic) and uncertainties (random).

*We checked and changed to the appropriate term accordingly (L55, 57, 62, 182, 261, 294, 417, 454, and 456).*

L83/84: Since 2006 IFS HRES has undergone several configuration changes, some of them affecting horizontal and vertical resolution. Please mention at which resolution

the input data was available (spatial and temporal) for STILT and if and when configurations changed.

> *Throughout the study period we have consistently extracted the IFS HRES data retrieved from ECMWF at 3-hourly temporal resolutions.*
>
> *The spatial resolution of HRES model was indeed changed two times throughout our study period: the original atmospheric model grid resolution of $T_L799$ was updated first in January 2010 (Cycle 36r1) T1279, and second time in March 2016 (Cycle 41r2) to O1280. 0.25° x 0.25° spatial grid that we have used throughout the study period is roughly to the original $T_L799$ configuration, so these two updates did not impact the quality of the meteorological fields more significantly than any other updates of the HRES system.*
>
> *More substantial change of the vertical resolution occurred on June 26th, 2013, with the introduction of Cycle 38r2. The configuration of the model levels changed that day from L91 to L137, and we have extracted the data at the higher vertical resolution since that time. Consequently, in STILT, vertical levels from 1 to 60 are used before June 26th, 2013 and from 1 to 90 after this date, always covering the atmosphere between the surface to altitude of approximately 20.1 km agl.*
>
> *As per suggestion, we've expanded the description in the revised manuscript (L106-111).*

L85: The number of released particles seems very small. Over which time interval are these particles released? Other inverse modelling studies using Lagrangian transport models employ much larger particle numbers (e.g. Lauvaux et al. 2016) even though their domains and transport times are smaller. If only 100 particles are released per hour of measurements but residence times are evaluated in a grid of approximately (160 x 160 = 25'600) cells, it seems unlikely to get a statistically robust estimate of residence times.

> *The particles are released at stations every hour following the continuous observations measured at hourly time intervals. Regarding the limited number of released particles, the horizontal size of footprint grid cells is dynamically adjusted (increased) according to the increase of footprints area when particles leave apart from the receptor. The reasons are: 1) to reduce the computation time, since it is proportional to the number of particles, and 2) to avoid under-sampling of the surface fluxes when the statistical probability becomes smaller to find a particle in a certain grid box as explained in a study conducted by Gerbig et al. (2003).*
>
> Additionally, error due to limited number of particles is fully random, and amounts to about 10% of the regional flux signal (i.e., about 1 ppm). When comparing this to the assumed model-data mismatch uncertainty (1.5 ppm for tall tower measurements over weekly aggregated measurements), and taking into account that there are 42 hourly measurements per week (6 per day), the impact of the random uncertainty from the relatively small number of particles is negligible.

> *The temporal resolution of footprints has been indicated in the revised manuscript, L104.*

L110: Depending on the height of the mountain, I would rather say that these sites experience free tropospheric conditions than residual layer conditions.
> *We have adjusted the text as per suggestion (L137-138).*

L145/146: What is this choice based on? Which values did these length scales take? Are these decay distances constant over the whole domain?
> *This choice (nBVH) investigated by Kountouris et al. (2018) was based on applying a hyperbolic spatial correlation decay instead of the exponential decay that needed to add a bias term to the biosphere model, so that the annually aggregated uncertainty should match the assumed prior uncertainty. Therefore, this is remedied by applying the hyperbolic correlation decay as no bias needed under this scenario (nBVH). The spatial correlation lengths are around 66 km in zonal and 33 km in meridional direction.*
> *This explanation has been adapted in the revised manuscript (L178-180).*

L150: Were the CarboScope ocean fluxes updated since the 2013 publication or were these also climatologies?
> *These fluxes are updated in the CarboScope global inversion based on dataset of the Surface Ocean CO2 Atlas pCO2 observations, so the Carboscope ocean fluxes comprise seasonal, interannual, and day-to-day variations, http://www.bgc-jena.mpg.de/CarboScope/?ID=oc_v2021. Added information in L187-188.*

L152: What is the benefit of using EDGAR 4.3 over the much more recent EDGAR 6.0, which provides more temporally resolved emissions up to 2018? Since these emissions are taken as 'truth', I wonder if a sensitivity inversion with alternative anthropogenic emissions should have been conducted. Would it be possible to make an estimate of how large an uncertainty may be added by the prescribed anthropogenic emissions?
> *The estimates that we used are indeed based on EDGAR 4.3, but are significantly expanded following the COFFEE approach to take into account year-to-year changes in fuel-type specific emissions based on BP statistics, which is not done in the base EDGAR dataset. The processing of EDGAR 6.0 would require significant effort and is only planned in the future at the moment. Our dataset was used and tested in numerous studies up to date (e.g. [REFERENCES]), also in cooperation with authors involved in development of EDGAR. This leads us to believe that the methodology of COFFEE is sound and that the emissions predicted for years past 2012 (base year of EDGAR 4.3) are accurate. We modified the text in L188-190 slightly to further clarify this.*
> *We would like to thank the reviewer for the suggestion of the sensitivity study. This has not yet been done, but we plan to use EDGAR 6.0 together with*

> *different emission products in the future estimates to evaluate the impact of emissions on estimating NEE in the inverse modelling. We also provided a simple estimate of emission uncertainty in L193-200 (as explained under the General comments).*

L165: The sample standard deviation when using only two or three samples is not a very robust estimator of the true standard deviation and is generally biased low. This should be stated as a warning when comparing these ensemble spreads with 'true' standard deviations as used in prior uncertainty.

> *We agree, this argument is true and we mentioned it in the revised manuscript (L237-238).*

L167/168: In Bayesian inversions, it is usually possible to calculate the posterior covariance directly. It sounds as if this is not the case here. The posterior uncertainty will also depend on the prescribed prior uncertainty. Here, it seems that the prior uncertainty is not considered but only the data-mismatch uncertainty is used. Please clarify.

> *Indeed, the posterior uncertainty is calculated based on both prior and model-data mismatch uncertainties, so we added that in the revised manuscript (L216-217).*

Section 3.1: Results in this section only span the period 2006-2018 and exclude 2019. Although, this is correctly listed in Table 2, this limitation is not mentioned in the text and the title of the manuscript suggests that results would include 2019. Please clarify in more detail that different periods are used and why.

> *We used the inversions B0, B1, and B2 to facilitate the reference to the biosphere ensemble that differ in the biosphere models, and the period of time in Section 3.1 was restricted to 2006-2018 based on the availability of SiBCASA fluxes (due to unavailability of meteorology data used to force the model) with which the time series of VPRM and FLUXCOM fluxes overlap. We clarified this in the revised version in L210-211.*
>
> *Therefore, the analysis in Section 3.1 is devoted to highlighting the spread and uncertainty resulting from the choice of using different inputs over the overlapping time of such inputs. In Section 3.2 we did use 2006-2019 inversions of which we confined the use of observations to the "core sites" that have consistent coverage of observations over long time within the period of interest (e.g. S1 inversion) as well as to "recent sites" (e.g. S2 inversion) to avoid annual variations resulting from gaps in measurements from year to year so as to analyse the anomalies and IAV of NEE over years in the context of climate variation, in particular in 2018 and 2019 in line with 2006-2017 period of time. The title therefore reflects the complete period of time used in all ensembles of inversions.*

**Final authors' response (AC) to the referee comments (RC2) on acp-2021-873**

Section 3.1: Results in this section are only presented for Central, Northern and all of Europe. It would be interesting to see the results for the other regions as well. Could similar figures as Fig 3 and 5 also be provided for the other regions as part of a supplement? Especially since there are some references in the text to other regions (e.g. L191).

*Figures similar to 3 and 5 have been provided in the supplementary for the other regions, South, West and East of Europe (Figures S1 – S2). Figure S1 is also mentioned in the revised manuscript (L243).*

Table 3: Provide results for all of Europe. How is it possible that spread reduction is 95.1 % across all of Europe if Central Europe has a spread reduction of 96% and all other regions will have considerable less reduction? Without seeing the results for all European regions, the discussion remains unsatisfactory.

*Results of the spread and its reduction for all of Europe have been included in Table 3.*

*The formula used to calculate the uncertainty reduction is (prior_spread – posterior_spread)/prior_spread, so the spread of a-posteriori and a-priori is computed for the aggregated fluxes over the underlying regions, as well as over the aggregated fluxes of the full domain. This implies that the uncertainty over the full domain is not calculated as the sum of uncertainties over subregions. Therefore, it is expected that anticorrelations in the annual variations over the underlying regions across the domain lead to different interannual variability over the full domain. This is the case in the a-posteriori fluxes, of which the variability is driven by atmospheric data, thus dependent on the distribution of atmospheric sites. Hence, the uncertainty reduction may differ from the view of subregions to the full domain as the spatial variability differs from region to region. This is more pronounced in the a-posteriori, where the spread of the full domain cannot simply be the sum of underlying region spreads. In turn, this might be true for the a-priori with which the total spread domain-wide can be approximated as the sum of underlying region spreads due to the fact that spatial variability of biosphere models is, to some extent, correlated over all regions. In any case, the uncertainty reduction is an indication of to which extent the atmospheric data derive the a-posteriori. Similarly, as an example, the reduction of Bayesian uncertainty in 2018 for all Europe and Central Europe was found to be 87.9 and 87.3%, respectively.*

L216: This is already discussed in L195, where a reference to table 3 (with the same numbers) is given. Please consolidate.

*This has been consolidated in the revised manuscript, L247.*

Figure 7, L242-247: How is it possible that the innovation over south-eastern Europe is almost as large as in Central Europe even though there are no observations in this area (B0, B1)? Should the posterior in that case not stay very close to the prior?

**Final authors' response (AC) to the referee comments (RC2) on acp-2021-873**

*This influence is inherited from the hyperbolic spatial correlation function that has a wider impact further away from sites, given the influence of footprints from stations located in Central Europe over south-eastern Europe. In addition, the largest overestimation of $CO_2$ uptake in biosphere models is seen over south-eastern Europe, particularly for VPRM and to a lesser degree for FLUXCOM, which increases the magnitude of corrections for this region. In contrast, this is not the case when comparing the innovations of SiBCASA to VPRM and FLUXCOM, due to the fact that SiBCASA does not show special pattern of fluxes over such a region.*

L249: This sounds as if a different inversion system as compared to the previous section was used. Please clarify.

*It is the same inversion system, but in this section, we used the inversions that cover the full period 2006-2019 such as S1 setup. In addition, we conducted another inversion run using "core sites" with FLUXCOM as a biosphere model to investigate seasonal variations of posterior estimates in line with their respective priors obtained from the VPRM and FLUXCOM models. We have further clarified this in the revised manuscript (L315-316).*

L250: It is the same starting time as in the previous section. Only the additional year 2019 seems different.

*Yes, but as was already mentioned in a pervious comment, we typically relied on inversions that use observations from core sites over 2006-2019 to avoid the misattribution of IAV originating from dataset gaps over the targeted years. In the inversions set-ups B0, B1, and B2, observations were assimilated from all sites available across the domain despite their coverage length.*

L259: 'Seasonal NEE': Did you mean summer? Otherwise, the described years with positive anomalies don't make sense.

*Yes, this what was meant, and have been corrected in the revised manuscript accordingly (L320).*

L274/275: The biospheric signal is generally weak in winter (as predicted by the biosphere parameterisations). The IAV seen in the posterior may also be attributable to IAV in fossil fuel emissions that are not well represented in the used inventory. Please comment.

*In comparison with the rest of seasons, biosphere signal is still notably weak in winter even in the posterior IAV, which has been confirmed in Fig.10, but of course not as weak as prior IAV. Being dependent on remote sensing data, VPRM and FLUXCOM are anticipated to underestimate NEE variability during winters due to less information retrieved from satellite. This missing signal in the biosphere is likely to be seen in the inverse modelling from the atmospheric data. It is also possible that misrepresentation of fossil fuel emissions*

> *contributes to such a variability; however, this cannot be verified without knowledge of the IAV of the true emissions.*
> *This has been clarified in the revised manuscript (L342-344).*

L283: Why was this temperature dataset selected over ERA-5, which was used to drive the FLUXCOM estimates?

> *Actually, there is no specific reason. The main motivation to use them was that this dataset is independent of what has been used in the biosphere models and in the regional transport model so as to avoid any systematicity in the correlations. It is also incorporated into the CarboScope inversion and has been used in a prior study by Rödenbeck et al. (2020).*

L288/289: I do not see this at all. The summer anomalies in earlier years (2007, 2010, 2012) were much more dramatic. Even if we discard these because of the poorer data coverage (but why show them then in the figure?), summer 2018 does not seem too exceptional. Wouldn't it make more sense to discuss the growing season as a whole instead of summer and spring separately? Spring 2018 and 2019 look more exceptional to me than the summer months. Or limit the discussion to the whole year as is done below.

> *Yes, we agree that it makes more sense to discuss the reduction of CO2 uptake during the growing season in the light of climate anomalies rather than the separate seasons, so this has been changed in the revised manuscript (L355-356).*

L312: Could this also be driven by a generally earlier start of the growing season (not just in evergreens) already towards the end of February? Especially in southern Europe many crops start developing around that time already. Finally, this may once again be a misattribution of anthropogenic emissions. Warmer winters mean less anthropogenic emissions, if these are not considered correctly or fully the "missing" CO2 may be attributed to biospheric uptake!

> *An earlier start of growing season can be associated with warm temperature which may explain part of the anticorrelations between T and NEE during warmer winters through the GPP; however, the length of time shift of growing season towards end of February is expected to be a few days and thus having less impact on winter fluxes. This would need a further study. From the carbon uptake length during thermal growing season, the shift in the onset and termination was about 4 days in Barichivich et al. (2012) conducted on the northern hemisphere. In addition, the stronger anticorrelation we observed over central Europe -0.83 versus -0.4 over southern Europe does not indicate a special role of southern Europe with regard to warmer winters. The impact of onset and termination of carbon uptake on winter NEE might be done in a separate study. On the other hand, higher NEE in colder winters can be attributed to increasing soil respiration in a warmer soil due to snow insulation.*

**Final authors' response (AC) to the referee comments (RC2) on acp-2021-873**

> *We agree that the interpretation of these anticorrelations holds true in case of misattribution of anthropogenic emissions, so we added this scenario as part of the discussion in the revised manuscript (L383-385).*

L321: 'identical observations'. Do you mean an identical set of observation sites? Otherwise, this could be misunderstood in the sense that identical observations were repeatedly used each year.

> *Yes, what we mean here is that the observational data collected from that set of sites and used in this inversion (S2) have no gaps over the respective years. We agree that 'identical observations' is confusing, so we rephrased it to "an identical set of observation sites" in the revised version (L393).*

L321: Why are none of the German ICOS sites used here? Were they not available in 2018 and 2019? Why restrict to only 16 sites when trying to analyse the spatial differences?

> *In fact, (as mentioned above) the selection of the subset of these sites used in S2 inversion is mainly restricted to sites that have a full coverage of data over the recent five years (2014-2019) regardless of their locations, in order to avoid any variations of NEE estimates that might be caused by data gaps. Therefore, S2 inversion is the best candidate to be used for comparing interannual variations of NEE within the last five years, as being done for 2018 and 2019 to highlight the climate variation impact through the difference between both years. The German ICOS sites had data gaps either in 2019 or before, which was the reason for their exclusion.*

L329-331: This is not correct. Over the UK we can see basically no temperature anomaly in summer 2019, a positive SPEI anomaly but still largely increased posterior NEE. Please be more precise in the description.

> *We corrected that in the revised manuscript (L403) by confining the comparison to Central Europe as a clear coincidence between posterior NEE, T and SPEI seen, given that Central Europe is the best example of well-constrained regions.*

L400: Most of western and central Europe do not experience any lasting snow cover anymore! Periods with snow cover are mostly limited to a few days. Colder winters also don't necessarily mean more snow as cold periods in central Europe are usually connected to easterly advection in high pressure systems with little precipitation. Overall, this 'theory' would need to be evaluated with additional datasets (snow cover, soil temperatures, etc.).

> *We agree, this theory needs to be evaluated in a further study with additional datasets of snow cover, soil temperature and air temperature at the continental scale. Nonetheless, our interpretation is supported by a study conducted by Monson et al. (2006) highlighting the impact of snow-depth on NEE during*

*winter using datasets of snow-depth, soil temperature and air temperature. They indicated that increasing depth of snow cover leads to increasing soil respiration, and vice versa. Therefore, this interpretation has been carefully readapted in the revised manuscript (L477) based on the available study.*

Figure 11: Why exclude fall here?

*We agree that we should add fall differences for a complete seasonal comparison, so differences in fall have been added in the revised manuscript. NEE estimates during fall of 2018 also suggests less uptake over western, northern and southern Europe compared with 2019. We have also slightly added this outcome in the revised manuscript (L397-398).*

Definition of winter season: Which months are incorporated into the winter estimate of a specific year (X)? Jan X, Feb X and Dec X, or Dec X, Jan X+1, Feb X+1? Please clarify. If the first definition is used then there is no connection in the climatological sense and the interpretation may be more difficult.

*Indeed, so the second definition is used in our analysis. This has been clarified in L385-386 in the revised manuscript.*

**4) Technical comments**

L38: NEE was only defined in abstract. Please redefine in main text.

*We have defined it again in the introduction in the revised manuscript (L40-41).*

L103: Remove line. Seems to be a mistake.

*Yes, that was a mistake. It has been removed in the revised manuscript (L130).*

---

## Author Response (AR1)

**Point-by-point response to the reviews on acp-2021-873**

*We are very thankful to the reviewers for the constructive comments and for considering the manuscript for publication in ACP after minor revisions, which we have done according to the suggestions of reviewers. In the following, we address the reviewers' comments under the respective sections — i.e., General, General comments, Detailed comments, Specific comments, Technical corrections, and Technical comments.*

*Note: The reviewer comments (RC1) are referred to in "Arial" font type throughout the texts, and the authors' responses are referred to as "Italic Arial" with indented lines.*

*Abbreviations used for the point-by-point response:*
***RC1***: *Anonymous Referee 1 comments*
***RC2***: *Anonymous Referee 2 comments*
***AC***: *Authors' comments*
***CM***: *Relevant Changes made to the revised manuscript by the Authors*

**1- Anonymous Referee #1 (RC1)**

**1) General comments**

**RC1:** Authors present an analysis of the European terrestrial carbon cycle variability in 2006-2019 made with the pre-operational inverse modelling framework "CarboScope Regional". The CO2 flux estimates are shown to be largely independent from the prior fluxes in the area of dense observations. The results confirm dominance of the observational constraint on fluxes and the importance of climate controls on the interannual flux variability. Authors find the inverse model predicts statistically significant positive CO2 flux anomalies in 2018-2019 related to hot and dry climate in those anomalous years. The paper is well written and can be considered for publication after minor revisions.

> *AC: Thank you for considering the paper to be published in the ACP journal. We have revised our manuscript accordingly as detailed below.*

**2) Detailed comments**

**RC1:** L75-80 Although some of the information can be found in references, to improve readability it is useful to give few more details about the CSR such as optimization scheme and temporal resolution of flux corrections.

> *AC: We agree that adding more information about the optimization scheme is useful, so a complementary description is added in the methods in the revised manuscript.*
> *CM: A complementary description is added in L81-97.*

**RC1:** L104 Need to give detail – where station types come from.

> *AC: The station types are categorized with different classes according to the ability for the regional transport model to reasonably simulate the atmospheric concentration, given the variable complexity to represent the local circulation, over each station as explained in Rödenbeck, (2005). We have added this information in the revised manuscript.*
> *CM: Additional detail is included in L125-127.*

**RC1:** L248-L300 The correlation of posterior fluxes with climate indices has been reported in detail. To enhance the validation of interannual flux variability, can authors add comparison with interannually varying regional flux estimates by independent process-based models, and possibly, top-down?

> *AC: We have added an additional comparison on interannual variability of flux estimates in the supplementary materials (subtitled as "validation test on posterior IAV", also Fig. S3) using an independent top-down model (LUMIA inversion) which serves as a validation of IAV in our results. The validation showed good agreement in the IAV between both estimates despite different data inputs and inversion setups between both inversion systems.*
> *CM: A description under the section of "validation test on posterior IAV" is provided in the supplementary materials (Fig. S3).*

**RC1:** L407 Need a reference here on systematic bias in transport models.

> *AC: Reference has been added.*
> *CM: A reference is added in L455.*

**3) Technical corrections**

**RC1:** L18 Phrase 'We further investigate the unprecedented increase of temperature …' is somewhat incomplete, better write that one investigates the impact of 'unprecedented increase ..' on the carbon cycle.

> *AC: This has been rephrased accordingly.*
> *CM: Rephrasing correction is made in L18-19.*

**RC1:** L103 'South-eastern Europe (light red).' Line out of place.

> *AC: This line is removed.*
> *CM: 'South-eastern Europe (light red).' is deleted from L125.*

**RC1:** L265 'fluxes of both' can be replaced with 'fluxes estimated with both'

> *AC: We changed it in the revised version.*
> *CM: The respective correction is made to L307.*

**RC1:** L405 'widespread scale' can be reduced to 'wide scale'

> *AC: It is changed based on the suggestion.*
> *CM: 'widespread scale' is reduced to 'wide scale' in L453.*

**Point-by-point response to the reviews on acp-2021-873**

**RC1:** L424-425 The phrase 'spatial correlation length of prior error' can be reformulated, it would be more accurate to avoid using 'prior' as this spatial correlation is applied to posterior flux corrections.

> *AC: The spatial correlation actually belongs to the prior error, together with the temporal error correlation it forms the assumed error structure associated with the prior uncertainty.*
>
> *CM: Rephrasing to ´a-priori spatial correlation length´ in L473-474.*

**RC1:** L460 Paper number in Chevallier 2012b is missing (Global Biogeochem. Cycles, 26, GB1021, doi:10.1029/2010GB003974)

> *AC: We corrected the reference information accordingly.*
>
> *CM: The missing paper number ´Artn Gb1021´ is corrected in L515.*

**2- Anonymous Referee #2 (RC2)**

**1) General**

**RC2:** The study presents inverse modelling estimates of the net ecosystem exchange across Europe for the period 2006 to 2019 with a special focus on the exceptional years 2018/2019 and the analysis of influence of environmental drivers (temperature, moisture availability) on NEE. The applied methods have been established in previous work and are generally sound. However, a few general questions concerning the method should be addressed (see comments below). The manuscript is well organized and generally well written. Nevertheless, in some cases the description of results could be more precise. Overall, I recommend the manuscript for publication after a series of rather minor issues have been addressed.

> *AC: Thank you for considering the paper to be published in the ACP journal. We have revised our manuscript accordingly as detailed below.*

**2) General comments**

**RC2:** Title: In the title and elsewhere the inversion system is labelled 'pre-operational'. However, there is no discussion whatsoever, why this term is used. Furthermore, if the current system is the pre-operational system, what do the authors envision as the operation system? Either this label has to be discussed in more detail or it should be removed from the text and title.

> *AC: Pre-operational means that the system is used to provide annual updates of estimated fluxes. It is not an operational system as the system is under development from year to year, we have clarified that in the revised manuscript.*
>
> *CM: Detail on the 'pre-operational' system is provided in L288-291.*

**Point-by-point response to the reviews on acp-2021-873**

**RC2:** Anthropogenic emissions: Anthropogenic CO2 emissions are assumed to be well-known for the current study (L51) and are not updated as part of the inversion. How sure can we be about this? While the sensitivity of the inversion towards prior biospheric fluxes and prescribed ocean fluxes is investigated, no such analysis is done for the anthropogenic fluxes. Some kind of discussion of this assumption should be included in the manuscript. What is the potential uncertainty added by this assumption? This, especially in the light of interannual variability and the timing of anthropogenic emissions that exhibit larger uncertainties than annual national totals.

> *AC: Evaluating emission uncertainty is still challenging as the truth of spatio-temporal emissions cannot not be easily reported. Nevertheless, it has been the standard approach in CO2 inversions to solve for the highly uncertain biospheric exchange fluxes and to assume that the anthropogenic emissions are well known (Rödenbeck et al., 2020; Peylin et al., 2013; Chevalier et al., 2012, Monteil et al., 2020).*
>
> *A simple estimate of the uncertainty associated with anthropogenic emissions can be made when comparing the recently updated fossil fuel emissions over EU27+UK in 2014 as reported in Petrescu et al. (2021) obtained from eight data sources: BP, EIA, CEDS, EDGAR, GCP, IEA, CDIAC, and NGHGI (UNFCCC, 2019), the spread between the annual total emissions is about 0.038 PgC (with a mean of 0.974 PgC). The prior and posterior uncertainty of NEE amounts to 0.490 and 0.037 PgC per year, respectively. Prior NEE uncertainty is by far dominating in the inversion in comparison with emission uncertainty, which is within the same order of magnitude of posterior NEE uncertainty. This implies, the uncertainty in emissions would be about 4% whatever emission products we prescribe in the inversion among those abovementioned data sources. As a result, prescribing fossil fuel emissions in the inversion and solving for NEE is appropriate. However, when interpreting posterior biosphere-atmosphere exchange fluxes one has to take into account that part of the fluxes and their variability might be compensating for errors in anthropogenic emissions. Future releases of the pre-operational system will include different anthropogenic emission estimates and assess the resulting uncertainty in a separate study with more detail, making use of the recent emission products estimated from - e.g., TNO and EDGAR_v6. This has been clarified in the revised manuscript.*
>
> *CM: A discussion on the assumption of fossil fuel emissions is expanded in L183-190.*

**3) Specific comments**

**RC2:** L44-46: Consider adding references to this statement.
> *AC: References added in the revised manuscript.*
> *CM: References are added to L47.*

**RC2:** L53 and elsewhere: The terms 'error' and 'uncertainty' seem to be used interchangeable throughout the manuscript. Consider using 'uncertainties' throughout

and distinguish between random and non-random uncertainties, if that was the intention when using error (systematic) and uncertainties (random).

> *AC: We checked and changed to the appropriate term accordingly.*
>
> *CM: `error(s)` is replaced with `uncertainty(-ies)` in L54, L56, L61, L172, L247, L278, L389, and L425.*

**RC2:** L83/84: Since 2006 IFS HRES has undergone several configuration changes, some of them affecting horizontal and vertical resolution. Please mention at which resolution the input data was available (spatial and temporal) for STILT and if and when configurations changed.

> *AC: Throughout the study period we have consistently extracted the IFS HRES data retrieved from ECMWF at 3-hourly temporal resolutions.*
>
> *The spatial resolution of HRES model was indeed changed two times throughout our study period: the original atmospheric model grid resolution of $T_L799$ was updated first in January 2010 (Cycle 36r1) T1279, and second time in March 2016 (Cycle 41r2) to O1280. 0.25° x 0.25° spatial grid that we have used throughout the study period is roughly to the original $T_L799$ configuration, so these two updates did not impact the quality of the meteorological fields more significantly than any other updates of the HRES system.*
>
> *More substantial change of the vertical resolution occurred on June 26th, 2013, with the introduction of Cycle 38r2. The configuration of the model levels changed that day from L91 to L137, and we have extracted the data at the higher vertical resolution since that time. Consequently, in STILT, vertical levels from 1 to 60 are used before June 26th, 2013 and from 1 to 90 after this date, always covering the atmosphere between the surface to altitude of approximately 20.1 km agl.*
>
> *As per suggestion, we've expanded the description in the revised manuscript.*
>
> *CM: A description on the configurations of ECMWF dataset is provided in L102-107.*

**RC2:** L85: The number of released particles seems very small. Over which time interval are these particles released? Other inverse modelling studies using Lagrangian transport models employ much larger particle numbers (e.g. Lauvaux et al. 2016) even though their domains and transport times are smaller. If only 100 particles are released per hour of measurements but residence times are evaluated in a grid of approximately (160 x 160 = 25'600) cells, it seems unlikely to get a statistically robust estimate of residence times.

> *AC: The particles are released at stations every hour following the continuous observations measured at hourly time intervals. Regarding the limited number of released particles, the horizontal size of footprint grid cells is dynamically adjusted (increased) according to the increase of footprints area when particles leave apart from the receptor. The reasons are: 1) to reduce the computation time, since it is proportional to the number of particles, and 2) to avoid under-sampling of the surface fluxes when the statistical probability becomes smaller*

**Point-by-point response to the reviews on acp-2021-873**

> *to find a particle in a certain grid box as explained in a study conducted by Gerbig et al. (2003).*
>
> Additionally, error due to limited number of particles is fully random, and amounts to about 10% of the regional flux signal (i.e., about 1 ppm). When comparing this to the assumed model-data mismatch uncertainty (1.5 ppm for tall tower measurements over weekly aggregated measurements), and taking into account that there are 42 hourly measurements per week (6 per day), the impact of the random uncertainty from the relatively small number of particles is negligible.
>
> *The temporal resolution of footprints has been indicated in the revised manuscript*
>
> **CM:** *´over hourly time intervals´ has been added in L100.*

**RC2:** L110: Depending on the height of the mountain, I would rather say that these sites experience free tropospheric conditions than residual layer conditions.

> **AC:** *We have adjusted the text as per suggestion in the revised version.*
>
> **CM:** *Sentence in L132-133 has been modified to ´as mountain sites experience free tropospheric conditions, depending on the mountain height´.*

**RC2:** L145/146: What is this choice based on? Which values did these length scales take? Are these decay distances constant over the whole domain?

> **AC:** *This choice (nBVH) investigated by Kountouris et al. (2018) was based on applying a hyperbolic spatial correlation decay instead of the exponential decay that needed to add a bias term to the biosphere model, so that the annually aggregated uncertainty should match the assumed prior uncertainty. Therefore, this is remedied by applying the hyperbolic correlation decay as no bias needed under this scenario (nBVH). The spatial correlation lengths are around 66 km in zonal and 33 km in meridional direction.*
>
> *This explanation has been adapted in the revised manuscript.*
>
> **CM:** *Detail on the spatial correlation decay is added in L168-170.*

**RC2:** L150: Were the CarboScope ocean fluxes updated since the 2013 publication or were these also climatologies?

> **AC:** *These fluxes are updated in the CarboScope global inversion based on dataset of the Surface Ocean CO2 Atlas pCO2 observations, so the Carboscope ocean fluxes comprise seasonal, interannual, and day-to-day variations, http://www.bgc-jena.mpg.de/CarboScope/?ID=oc_v2021.*
>
> **CM:** *Additional information on CarboScope ocean fluxes is provided in L177-178*

**RC2:** L152: What is the benefit of using EDGAR 4.3 over the much more recent EDGAR 6.0, which provides more temporally resolved emissions up to 2018? Since these emissions are taken as 'truth', I wonder if a sensitivity inversion with alternative anthropogenic emissions should have been conducted. Would it be possible to make

an estimate of how large an uncertainty may be added by the prescribed anthropogenic emissions?

> *AC: The estimates that we used are indeed based on EDGAR 4.3, but are significantly expanded following the COFFEE approach to take into account year-to-year changes in fuel-type specific emissions based on BP statistics, which is not done in the base EDGAR dataset. The processing of EDGAR 6.0 would require significant effort and is only planned in the future at the moment. Our dataset was used and tested in numerous studies up to date (e.g. [REFERENCES]), also in cooperation with authors involved in development of EDGAR. This leads us to believe that the methodology of COFFEE is sound and that the emissions predicted for years past 2012 (base year of EDGAR 4.3) are accurate. We modified the text in L178-180 slightly to further clarify this.*
>
> *We would like to thank the reviewer for the suggestion of the sensitivity study. This has not yet been done, but we plan to use EDGAR 6.0 together with different emission products in the future estimates to evaluate the impact of emissions on estimating NEE in the inverse modelling. We also provided a simple estimate of emission uncertainty in the revised manuscript (as explained under the General comments).*
>
> *CM: Slight modifications are made to L178-180.*

**RC2:** L165: The sample standard deviation when using only two or three samples is not a very robust estimator of the true standard deviation and is generally biased low. This should be stated as a warning when comparing these ensemble spreads with 'true' standard deviations as used in prior uncertainty.

> *AC: We agree, this argument is true and we mentioned it in the revised manuscript.*
>
> *CM: The sentence ´Note that calculating the spread over a small size of samples might not reflect the true standard deviation´ is added to L223-224.*

**RC2:** AL167/168: In Bayesian inversions, it is usually possible to calculate the posterior covariance directly. It sounds as if this is not the case here. The posterior uncertainty will also depend on the prescribed prior uncertainty. Here, it seems that the prior uncertainty is not considered but only the data-mismatch uncertainty is used. Please clarify.

> *AC: Indeed, the posterior uncertainty is calculated based on both prior and model-data mismatch uncertainties, so we added that in the revised manuscript.*
>
> *CM: ´prescribed prior uncertainty´ is added in L204, with slight modifications in the sentence (L203-204)*

**RC2:** Section 3.1: Results in this section only span the period 2006-2018 and exclude 2019. Although, this is correctly listed in Table 2, this limitation is not mentioned in the text and the title of the manuscript suggests that results would include 2019. Please clarify in more detail that different periods are used and why.

**Point-by-point response to the reviews on acp-2021-873**

> *AC: We used the inversions B0, B1, and B2 to facilitate the reference to the biosphere ensemble that differ in the biosphere models, and the period of time in Section 3.1 was restricted to 2006-2018 based on the availability of SiBCASA fluxes (due to unavailability of meteorology data used to force the model) with which the time series of VPRM and FLUXCOM fluxes overlap. We clarified this in the revised version in L197-198.*
>
> *Therefore, the analysis in Section 3.1 is devoted to highlighting the spread and uncertainty resulting from the choice of using different inputs over the overlapping time of such inputs. In Section 3.2 we did use 2006-2019 inversions of which we confined the use of observations to the "core sites" that have consistent coverage of observations over long time within the period of interest (e.g. S1 inversion) as well as to "recent sites" (e.g. S2 inversion) to avoid annual variations resulting from gaps in measurements from year to year so as to analyse the anomalies and IAV of NEE over years in the context of climate variation, in particular in 2018 and 2019 in line with 2006-2017 period of time. The title therefore reflects the complete period of time used in all ensembles of inversions.*
>
> *CM: A clarification of the overlapping time between the inversions of the biosphere ensemble is appended in L97-198.*

**RC2:** Section 3.1: Results in this section are only presented for Central, Northern and all of Europe. It would be interesting to see the results for the other regions as well. Could similar figures as Fig 3 and 5 also be provided for the other regions as part of a supplement? Especially since there are some references in the text to other regions (e.g. L191).

> *AC: Figures similar to 3 and 5 have been provided in the supplementary for the other regions, South, West and East of Europe (Figures S1 – S2). Figure S1 is also mentioned in the revised manuscript.*
>
> *CM: ´Fig. S1 in the supplementary´ is added in L229. Figures S1-2 are also included as part of the supplementary.*

**RC2:** Table 3: Provide results for all of Europe. How is it possible that spread reduction is 95.1 % across all of Europe if Central Europe has a spread reduction of 96% and all other regions will have considerable less reduction? Without seeing the results for all European regions, the discussion remains unsatisfactory.

> *AC: Results of the spread and its reduction for all of Europe have been included in Table 3.*
>
> *The formula used to calculate the uncertainty reduction is (prior_spread – posterior_spread)/prior_spread, so the spread of a-posteriori and a-priori is computed for the aggregated fluxes over the underlying regions, as well as over the aggregated fluxes of the full domain. This implies that the uncertainty over the full domain is not calculated as the sum of uncertainties over subregions. Therefore, it is expected that anticorrelations in the annual variations over the underlying regions across the domain lead to different interannual variability*

*over the full domain. This is the case in the a-posteriori fluxes, of which the variability is driven by atmospheric data, thus dependent on the distribution of atmospheric sites. Hence, the uncertainty reduction may differ from the view of subregions to the full domain as the spatial variability differs from region to region. This is more pronounced in the a-posteriori, where the spread of the full domain cannot simply be the sum of underlying region spreads. In turn, this might be true for the a-priori with which the total spread domain-wide can be approximated as the sum of underlying region spreads due to the fact that spatial variability of biosphere models is, to some extent, correlated over all regions. In any case, the uncertainty reduction is an indication of to which extent the atmospheric data derive the a-posteriori. Similarly, as an example, the reduction of Bayesian uncertainty in 2018 for all Europe and Central Europe was found to be 87.9 and 87.3%, respectively.*
*CM: Three rows are appended in Table 3.*

**RC2:** L216: This is already discussed in L195, where a reference to table 3 (with the same numbers) is given. Please consolidate.
*AC: This has been consolidated in the revised manuscript.*
*CM: ´(95.1%, 96.0%, and 74.8%, respectively)´ is moved from L255 to L233.*

**RC2:** Figure 7, L242-247: How is it possible that the innovation over south-eastern Europe is almost as large as in Central Europe even though there are no observations in this area (B0, B1)? Should the posterior in that case not stay very close to the prior?
*AC: This influence is inherited from the hyperbolic spatial correlation function that has a wider impact further away from sites, given the influence of footprints from stations located in Central Europe over south-eastern Europe. In addition, the largest overestimation of $CO_2$ uptake in biosphere models is seen over south-eastern Europe, particularly for VPRM and to a lesser degree for FLUXCOM, which increases the magnitude of corrections for this region.* In contrast, this is not the case when comparing the innovations of SiBCASA to VPRM and FLUXCOM, due to the fact that SiBCASA does not show special pattern of fluxes over such a region.

**RC2:** L249: This sounds as if a different inversion system as compared to the previous section was used. Please clarify.
*AC: It is the same inversion system, but in this section, we used the inversions that cover the full period 2006-2019 such as S1 setup. In addition, we conducted another inversion run using "core sites" with FLUXCOM as a biosphere model to investigate seasonal variations of posterior estimates in line with their respective priors obtained from the VPRM and FLUXCOM models. We have further clarified this in the revised manuscript.*
*CM: Additional clarification is added to L296-297 as ´1) S1 inversion set-up and 2) a similar set-up to S1 performed with FLUXCOM instead of VPRM´.*

**Point-by-point response to the reviews on acp-2021-873**

**RC2:** L250: It is the same starting time as in the previous section. Only the additional year 2019 seems different.

> *AC: Yes, but as was already mentioned in a pervious comment, we typically relied on inversions that use observations from core sites over 2006-2019 to avoid the misattribution of IAV originating from dataset gaps over the targeted years. In the inversions set-ups B0, B1, and B2, observations were assimilated from all sites available across the domain despite their coverage length.*

**RC2:** L259: 'Seasonal NEE': Did you mean summer? Otherwise, the described years with positive anomalies don't make sense.

> *AC: Yes, this what was meant, and have been corrected in the revised manuscript accordingly.*
> *CM: 'Seasonal´ is replaced with `Summer` in L301.*

**RC2:** L274/275: The biospheric signal is generally weak in winter (as predicted by the biosphere parameterisations). The IAV seen in the posterior may also be attributable to IAV in fossil fuel emissions that are not well represented in the used inventory. Please comment.

> *AC: In comparison with the rest of seasons, biosphere signal is still notably weak in winter even in the posterior IAV, which has been confirmed in Fig.10, but of course not as weak as prior IAV. Being dependent on remote sensing data, VPRM and FLUXCOM are anticipated to underestimate NEE variability during winters due to less information retrieved from satellite. This missing signal in the biosphere is likely to be seen in the inverse modelling from the atmospheric data. It is also possible that misrepresentation of fossil fuel emissions contributes to such a variability; however, this cannot be verified without knowledge of the IAV of the true emissions.*
> *This has been clarified in the revised manuscript.*
> *CM: A clarification of fossil fuel IAV impact on posterior NEE IAV has been expended in L319-321.*

**RC2:** L283: Why was this temperature dataset selected over ERA-5, which was used to drive the FLUXCOM estimates?

> *AC: Actually, there is no specific reason. The main motivation to use them was that this dataset is independent of what has been used in the biosphere models and in the regional transport model so as to avoid any systematicity in the correlations. It is also incorporated into the CarboScope inversion and has been used in a prior study by Rödenbeck et al. (2020).*

**RC2:** L288/289: I do not see this at all. The summer anomalies in earlier years (2007, 2010, 2012) were much more dramatic. Even if we discard these because of the poorer data coverage (but why show them then in the figure?), summer 2018 does not seem too exceptional. Wouldn't it make more sense to discuss the growing season as a whole instead of summer and spring separately? Spring 2018 and 2019 look more

exceptional to me than the summer months. Or limit the discussion to the whole year as is done below.

> *AC: Yes, we agree that it makes more sense to discuss the reduction of CO2 uptake during the growing season in the light of climate anomalies rather than the separate seasons, so this has been changed in the revised manuscript.*
>
> *CM: In L332-333, ´summers´ is replaced with ´during the growing season´ and ´positive NEE anomalies´ and ´exceptional and´ are removed from the texts in L333.*

**RC2:** L312: Could this also be driven by a generally earlier start of the growing season (not just in evergreens) already towards the end of February? Especially in southern Europe many crops start developing around that time already. Finally, this may once again be a misattribution of anthropogenic emissions. Warmer winters mean less anthropogenic emissions, if these are not considered correctly or fully the "missing" CO2 may be attributed to biospheric uptake!

> *AC: An earlier start of growing season can be associated with warm temperature which may explain part of the anticorrelations between T and NEE during warmer winters through the GPP; however, the length of time shift of growing season towards end of February is expected to be a few days and thus having less impact on winter fluxes. This would need a further study. From the carbon uptake length during thermal growing season, the shift in the onset and termination was about 4 days in Barichivich et al. (2012) conducted on the northern hemisphere. In addition, the stronger anticorrelation we observed over central Europe -0.83 versus -0.4 over southern Europe does not indicate a special role of southern Europe with regard to warmer winters. The impact of onset and termination of carbon uptake on winter NEE might be done in a separate study. On the other hand, higher NEE in colder winters can be attributed to increasing soil respiration in a warmer soil due to snow insulation. We agree that the interpretation of these anticorrelations holds true in case of misattribution of anthropogenic emissions, so we added this scenario as part of the discussion in the revised manuscript.*
>
> *CM: A few details on the scenario of anthropogenic emission impact on NEE IAV during winter have been provided to L358-360.*

**RC2:** L321: 'identical observations'. Do you mean an identical set of observation sites? Otherwise, this could be misunderstood in the sense that identical observations were repeatedly used each year.

> *AC: Yes, what we mean here is that the observational data collected from that set of sites and used in this inversion (S2) have no gaps over the respective years. We agree that 'identical observations' is confusing, so we rephrased it in the revised version.*
>
> *CM: In L368, ´identical observations´ is replaced by ´an identical set of observation sites´.*

**Point-by-point response to the reviews on acp-2021-873**

**RC2:** L321: Why are none of the German ICOS sites used here? Were they not available in 2018 and 2019? Why restrict to only 16 sites when trying to analyse the spatial differences?

> *AC: In fact, (as mentioned above) the selection of the subset of these sites used in S2 inversion is mainly restricted to sites that have a full coverage of data over the recent five years (2014-2019) regardless of their locations, in order to avoid any variations of NEE estimates that might be caused by data gaps. Therefore, S2 inversion is the best candidate to be used for comparing interannual variations of NEE within the last five years, as being done for 2018 and 2019 to highlight the climate variation impact through the difference between both years. The German ICOS sites had data gaps either in 2019 or before, which was the reason for their exclusion.*

**RC2:** L329-331: This is not correct. Over the UK we can see basically no temperature anomaly in summer 2019, a positive SPEI anomaly but still largely increased posterior NEE. Please be more precise in the description.

> *AC: We corrected that in the revised manuscript by confining the comparison to Central Europe as a clear coincidence between posterior NEE, T and SPEI seen, given that Central Europe is the best example of well-constrained regions.*
>
> *CM: ´and in the United Kingdom (if we ignore the anomalies over Spain due to absence of observations)´ is deleted from L378.*

**RC2:** L400: Most of western and central Europe do not experience any lasting snow cover anymore! Periods with snow cover are mostly limited to a few days. Colder winters also don't necessarily mean more snow as cold periods in central Europe are usually connected to easterly advection in high pressure systems with little precipitation. Overall, this 'theory' would need to be evaluated with additional datasets (snow cover, soil temperatures, etc.).

> *AC: We agree, this theory needs to be evaluated in a further study with additional datasets of snow cover, soil temperature and air temperature at the continental scale. Nonetheless, our interpretation is supported by a study conducted by Monson et al. (2006) highlighting the impact of snow-depth on NEE during winter using datasets of snow-depth, soil temperature and air temperature. They indicated that increasing depth of snow cover leads to increasing soil respiration, and vice versa. Therefore, this interpretation has been carefully readapted in the revised manuscript based on the available study.*
>
> *CM: Sentence in L448 is grammatically modified: We replaced ´implies´ with ´may imply´ and ´when´ with ´if´.*

**RC2:** Figure 11: Why exclude fall here?

> *AC: We agree that we should add fall differences for a complete seasonal comparison, so differences in fall have been added in the revised manuscript.*

*NEE estimates during fall of 2018 also suggests less uptake over western, northern and southern Europe compared with 2019. We have also slightly added this outcome in the revised manuscript.*

***CM:*** *Figure 11 is replaced by a new Figure that consists of fall differences. Additional information about fall estimates has also been provided in L371-372.*

**RC2:** Definition of winter season: Which months are incorporated into the winter estimate of a specific year (X)? Jan X, Feb X and Dec X, or Dec X, Jan X+1, Feb X+1? Please clarify. If the first definition is used then there is no connection in the climatological sense and the interpretation may be more difficult.

***AC:*** *Indeed, so the second definition is used in our analysis. This has been clarified in the revised manuscript.*

***CM:*** *The definition of winter months in our study is clarified in a new sentence in L360-361 as ´Of note, December X, January X+1 and February X+1 are incorporated into the winter estimate of a specific year (X)´.*

**4) Technical comments**

**RC2:** L38: NEE was only defined in abstract. Please redefine in main text.

***AC:*** *We have defined it again in the introduction in the revised manuscript.*

***CM:*** *´(Net Ecosystem Exchange )´ is added to L39-40.*

**RC2:** L103: Remove line. Seems to be a mistake.

***AC:*** *Yes, that was a mistake. It has been removed in the revised manuscript.*

***CM:*** *´South-eastern Europe (light red).´ is deleted from L125.*

---

## Referee Report (RR1)

I would like to thank the authors for carefully replying to comments raised by both referees. The replies are appropriate and clarifications have been added to revised manuscript. There are only some very minor comments remaining, concerning the newly added description of the CSR system. Once these are adjusted, the study is fit for publication.

Lines numbers as taken from Track-changes version of revised manuscript.

L86ff: The description seems overly complicated. Consider:

*"A global forward run is then carried out using 'global' observations to obtain simulated concentrations for the regional sites. A second forward run is conducted applying zero-fluxes outside of the regional domain. This can be considered as a regional run utilizing a global transport model at a coarse spatial resolution. The subtraction of the 'regional' run from the 'global' run results in the far field contribution on the sites within the regional domain. Subtracting the latter from the measurements and yields the remaining regional mixing ratio that is used in the regional inversion applying the regional-scale transport model at finer spatial resolution."*

Equation 1: To me it is not clear what the matrix **p** contains. Is this equivalent to the frequently used prior contribution to the cost function: $(x - x\_prior)^T R^{-1} (x-x\_prior)$? x being the state vector, x_prior the prior state and R the prior covariance matrix. Why would parameters p have zero mean? Please also add the variable name for the cost function in the text (L94).

Equation 2: I assume that Qc does not only contain observation uncertainty but also representativeness uncertainty?

---

## Author Response (AR2)

**Point-by-point response to the Editor review**

*We are very thankful to the Editor and the Anonymous Referee #2 for the constructive comments that helped improve the manuscript, as well as for the suggestions for the publication in ACP after a minor revision, which we have done accordingly.*
*In the following, we address the Editor and the Referee's comments.*

*Note: The Editor and Referee's comments (EC and RC2) are referred to in "Arial" font type throughout the texts, and the authors' responses are referred to as "Italic Arial" with indented lines.*

**Editor Comments (EC)**

**Comments to the author**:

Dear authors, your manuscript is basically accepted. Only two points are suggested to modify: I copy the following lines from the review. The line numbers are taken from the track changes version of the revised manuscript.

Kind regards

Mathias Palm

> *Thank you very much Mathias. We address these points under their respective texts in the following:*

L86ff: The description seems overly complicated. Consider:

"A global forward run is then carried out using 'global' observations to obtain simulated concentrations for the regional sites. A second forward run is conducted applying zero-fluxes outside of the regional domain. This can be considered as a regional run utilizing a global transport model at a coarse spatial resolution. The subtraction of the 'regional' run from the 'global' run results in the far field contribution on the sites within the regional domain. Subtracting the latter from the measurements and yields the remaining regional mixing ratio that is used in the regional inversion applying the regional-scale transport model at finer spatial resolution."

> *Thank you for the suggestion. We have modified this paragraph in the revised manuscript accordingly (L84-90).*

Equation 1: To me it is not clear what the matrix p contains. Is this equivalent to the frequently used prior contribution to the cost function: $(x - x\_prior)^T R^{-1} (x\text{-}x\_prior)$? x being the state vector, x_prior the prior state and R the prior covariance matrix. Why

would parameters p have zero mean? Please also add the variable name for the cost function in the text (L94).

> *In fact, in CSR the a-priori probability distribution of the fluxes is defined differently but equivalent to the traditional way indicated as $(x - x_{prior})^T R^{-1} (x - x_{prior})$ in which the covariance matrix is explicitly defined. In our system, the prior flux information is expressed in a linear flux model $f = f_{fix} + \boldsymbol{F} \times \boldsymbol{p}$ that comprises all flux components with a fixed term $f_{fix}$. The second term is an adjustable term that contains the spatio-temporal shape of fluxes in the matrix F and the adjustable parameters in the vector p. Information of the uncertainty and correlations of the a-priori are also contained in the matrix F. This construction allows for splitting flux model into different components (physical processes) and some of these processes can further be split into a different time scales which can be treated in the flux model as a separate component with different shapes. The mathematical description of the system scheme is explained in detail in Rödenbeck (2005).*
>
> *In L94-97 (in the revised manuscript) we added the Equation (2) $f = f_{fix} + \boldsymbol{F} \times \boldsymbol{p}$ and its variable definitions that can help the reader differentiate between the matrix that involves the flux uncertainty and the vector of adjustable parameters mentioned in Eq.(1). We have also referred the reader to Rödenbeck (2005) for more details about the mathematical scheme of the system (L102-103).*
>
> *The variable name of the cost function J is also added to L91 in the revised manuscript.*

Equation 2: I assume that Qc does not only contain observation uncertainty but also representativeness uncertainty?

> *Indeed, the Qc matrix contains the transport and representation uncertainty, as well as the measurement uncertainty. This has also been clarified in the revised manuscript (L99).*

**Anonymous Referee #2 (RC2)**

I would like to thank the authors for carefully replying to comments raised by both referees. The replies are appropriate and clarifications have been added to revised manuscript. There are only some very minor comments remaining, concerning the

**Point-by-point response to the Editor review**

newly added description of the CSR system. Once these are adjusted, the study is fit for publication.

Lines numbers as taken from Track-changes version of revised manuscript.

*Thank you very much for the constructive comments that help improve the manuscript.*
*Please find your comments addressed under the Editor section above.*